# Contextual semibandits via supervised learning oracles

**Akshay Krishnamurthy**[†]
akshay@cs.umass.edu

**Alekh Agarwal**[‡]
alekha@microsoft.com

**Miroslav Dudík**[‡]
mdudik@microsoft.com

[†]College of Information and Computer Sciences
University of Massachusetts, Amherst, MA

[‡]Microsoft Research
New York, NY

## Abstract

We study an online decision making problem where on each round a learner chooses a list of items based on some side information, receives a scalar feedback value for each individual item, and a reward that is linearly related to this feedback. These problems, known as contextual semibandits, arise in crowdsourcing, recommendation, and many other domains. This paper reduces contextual semibandits to supervised learning, allowing us to leverage powerful supervised learning methods in this partial-feedback setting. Our first reduction applies when the mapping from feedback to reward is known and leads to a computationally efficient algorithm with near-optimal regret. We show that this algorithm outperforms state-of-the-art approaches on real-world learning-to-rank datasets, demonstrating the advantage of oracle-based algorithms. Our second reduction applies to the previously unstudied setting when the linear mapping from feedback to reward is unknown. Our regret guarantees are superior to prior techniques that ignore the feedback.

## 1 Introduction

Decision making with partial feedback, motivated by applications including personalized medicine [21] and content recommendation [16], is receiving increasing attention from the machine learning community. These problems are formally modeled as *learning from bandit feedback*, where a learner repeatedly takes an action and observes a reward for the action, with the goal of maximizing reward. While bandit learning captures many problems of interest, several applications have additional structure: the action is combinatorial in nature and more detailed feedback is provided. For example, in internet applications, we often recommend sets of items and record information about the user's interaction with each individual item (e.g., click). This additional feedback is unhelpful unless it relates to the overall reward (e.g., number of clicks), and, as in previous work, we assume a linear relationship. This interaction is known as the *semibandit* feedback model.

Typical bandit and semibandit algorithms achieve reward that is competitive with the single best fixed action, i.e., the best medical treatment or the most popular news article for everyone. This is often inadequate for recommendation applications: while the most popular articles may get some clicks, personalizing content to the users is much more effective. A better strategy is therefore to leverage contextual information to learn a rich policy for selecting actions, and we model this as *contextual semibandits*. In this setting, the learner repeatedly observes a context (user features), chooses a composite action (list of articles), which is an ordered tuple of simple actions, and receives reward for the composite action (number of clicks), but also feedback about each simple action (click). The goal of the learner is to find a policy for mapping contexts to composite actions that achieves high reward.

We typically consider policies in a large but constrained class, for example, linear learners or tree ensembles. Such a class enables us to learn an expressive policy, but introduces a computational challenge of finding a good policy without direct enumeration. We build on the supervised learning literature, which has developed fast algorithms for such policy classes, including logistic regression and SVMs for linear classifiers and boosting for tree ensembles. We access the policy class exclusively through a supervised learning algorithm, viewed as an oracle.

| Algorithm | Regret | Oracle Calls | Weights $w^\star$ |
|---|---|---|---|
| VCEE (Thm. 1) | $\sqrt{KLT\log N}$ | $T^{3/2}\sqrt{K/(L\log N)}$ | known |
| $\epsilon$-Greedy (Thm. 3) | $(LT)^{2/3}(K\log N)^{1/3}$ | 1 | known |
| Kale et al. [12] | $\sqrt{KLT\log N}$ | not oracle-based | known |
| EELS (Thm. 2) | $(LT)^{2/3}(K\log N)^{1/3}$ | 1 | unknown |
| Agarwal et al. [1] | $L\sqrt{K^L T\log N}$ | $\sqrt{K^L T/\log N}$ | unknown |
| Swaminathan et al. [22] | $L^{4/3}T^{2/3}(K\log N)^{1/3}$ | 1 | unknown |

Table 1: Comparison of contextual semibandit algorithms for arbitrary policy classes, assuming all rankings are valid composite actions. The reward is semibandit feedback weighted according to $w^\star$. For known weights, we consider $w^\star = \mathbf{1}$; for unknown weights, we assume $\|w^\star\|_2 \leq O(\sqrt{L})$.

In this paper, we develop and evaluate oracle-based algorithms for the contextual semibandits problem. We make the following contributions:

1. In the more common setting where the linear function relating the semibandit feedback to the reward is known, we develop a new algorithm, called VCEE, that extends the oracle-based contextual bandit algorithm of Agarwal et al. [1]. We show that VCEE enjoys a regret bound between $\tilde{O}\big(\sqrt{KLT\log N}\big)$ and $\tilde{O}\big(L\sqrt{KT\log N}\big)$, depending on the combinatorial structure of the problem, when there are $T$ rounds of interaction, $K$ simple actions, $N$ policies, and composite actions have length $L$.[1] VCEE can handle structured action spaces and makes $\tilde{O}(T^{3/2})$ calls to the supervised learning oracle.

2. We empirically evaluate this algorithm on two large-scale learning-to-rank datasets and compare with other contextual semibandit approaches. These experiments comprehensively demonstrate that effective exploration over a rich policy class can lead to significantly better performance than existing approaches. To our knowledge, this is the first thorough experimental evaluation of not only oracle-based semibandit methods, but of oracle-based contextual bandits as well.

3. When the linear function relating the feedback to the reward is unknown, we develop a new algorithm called EELS. Our algorithm first learns the linear function by uniform exploration and then, adaptively, switches to act according to an empirically optimal policy. We prove an $\tilde{O}\big((LT)^{2/3}(K\log N)^{1/3}\big)$ regret bound by analyzing when to switch. We are not aware of other computationally efficient procedures with a matching or better regret bound for this setting.

See Table 1 for a comparison of our results with existing applicable bounds.

**Related work.** There is a growing body of work on combinatorial bandit optimization [2, 4] with considerable attention on semibandit feedback [6, 10, 12, 13, 19]. The majority of this research focuses on the non-contextual setting with a known relationship between semibandit feedback and reward, and a typical algorithm here achieves an $\tilde{O}(\sqrt{KLT})$ regret against the best fixed composite action. To our knowledge, only the work of Kale et al. [12] and Qin et al. [19] considers the contextual setting, again with known relationship. The former generalizes the Exp4 algorithm [3] to semibandits, and achieves $\tilde{O}(\sqrt{KLT})$ regret,[2] but requires explicit enumeration of the policies. The latter generalizes the LinUCB algorithm of Chu et al. [7] to semibandits, assuming that the simple action feedback is linearly related to the context. This differs from our setting: we make no assumptions about the simple action feedback. In our experiments, we compare VCEE against this LinUCB-style algorithm and demonstrate substantial improvements.

We are not aware of attempts to learn a relationship between the overall reward and the feedback on simple actions as we do with EELS. While EELS uses least squares, as in LinUCB-style approaches, it does so *without* assumptions on the semibandit feedback. Crucially, the covariates for its least squares problem are observed *after predicting a composite action* and not before, unlike in LinUCB.

Supervised learning oracles have been used as a computational primitive in many settings including active learning [11], contextual bandits [1, 9, 20, 23], and structured prediction [8].

## 2 Preliminaries

Let $\mathcal{X}$ be a space of contexts and $\mathcal{A}$ a set of $K$ simple actions. Let $\Pi \subseteq (\mathcal{X} \to \mathcal{A}^L)$ be a finite set of policies, $|\Pi| = N$, mapping contexts to composite actions. Composite actions, also called rankings, are tuples of $L$ distinct simple actions. In general, there are $K!/(K-L)!$ possible rankings, but they might not be valid in all contexts. The set of valid rankings for a context $x$ is defined implicitly through the policy class as $\{\pi(x)\}_{\pi \in \Pi}$.

Let $\Delta(\Pi)$ be the set of distributions over policies, and $\Delta_{\leq}(\Pi)$ be the set of non-negative weight vectors over policies, summing to at most 1, which we call subdistributions. Let $\mathbf{1}(\cdot)$ be the 0/1 indicator equal to 1 if its argument is true and 0 otherwise.

In stochastic contextual semibandits, there is an unknown distribution $\mathcal{D}$ over triples $(x, y, \xi)$, where $x$ is a context, $y \in [0,1]^K$ is the vector of *reward features*, with entries indexed by simple actions as $y(a)$, and $\xi \in [-1, 1]$ is the reward noise, $\mathbb{E}[\xi|x, y] = 0$. Given $y \in \mathbb{R}^K$ and $A = (a_1, \dots, a_L) \in \mathcal{A}^L$, we write $y(A) \in \mathbb{R}^L$ for the vector with entries $y(a_\ell)$. The learner plays a $T$-round game. In each round, nature draws $(x_t, y_t, \xi_t) \sim \mathcal{D}$ and reveals the context $x_t$. The learner selects a valid ranking $A_t = (a_{t,1}, a_{t,2}, \dots, a_{t,L})$ and gets reward $r_t(A_t) = \sum_{\ell=1}^L w_\ell^\star y_t(a_{t,\ell}) + \xi_t$, where $w^\star \in \mathbb{R}^L$ is a possibly unknown but fixed weight vector. The learner is shown the reward $r_t(A_t)$ and the vector of reward features for the chosen simple actions $y_t(A_t)$, jointly referred to as *semibandit feedback*.

The goal is to achieve cumulative reward competitive with all $\pi \in \Pi$. For a policy $\pi$, let $\mathcal{R}(\pi) := \mathbb{E}_{(x,y,\xi) \sim \mathcal{D}}\big[r\big(\pi(x)\big)\big]$ denote its expected reward, and let $\pi^\star := \operatorname{argmax}_{\pi \in \Pi} \mathcal{R}(\pi)$ be the maximizer of expected reward. We measure performance of an algorithm via cumulative empirical regret,

$$\text{Regret} := \sum_{t=1}^T r_t(\pi^\star(x_t)) - r_t(A_t). \tag{1}$$

The performance of a policy $\pi$ is measured by its expected regret, $\text{Reg}(\pi) := \mathcal{R}(\pi^\star) - \mathcal{R}(\pi)$.

**Example 1.** In personalized search, a learning system repeatedly responds to queries with rankings of search items. This is a contextual semibandit problem where the query and user features form the context, the simple actions are search items, and the composite actions are their lists. The semibandit feedback is whether the user clicked on each item, while the reward may be the *click-based discounted cumulative gain* (DCG), which is a weighted sum of clicks, with position-dependent weights. We want to map contexts to rankings to maximize DCG and achieve a low regret.

We assume that our algorithms have access to a *supervised learning oracle*, also called an *argmax oracle*, denoted AMO, that can find a policy with the maximum empirical reward on any appropriate dataset. Specifically, given a dataset $D = \{x_i, y_i, v_i\}_{i=1}^n$ of contexts $x_i$, reward feature vectors $y_i \in \mathbb{R}^K$ with rewards *for all simple actions*, and weight vectors $v_i \in \mathbb{R}^L$, the oracle computes

$$\text{AMO}(D) := \operatorname*{argmax}_{\pi \in \Pi} \sum_{i=1}^n \langle v_i, y_i(\pi(x_i)) \rangle = \operatorname*{argmax}_{\pi \in \Pi} \sum_{i=1}^n \sum_{\ell=1}^L v_{i,\ell} y_i(\pi(x_i)_\ell), \tag{2}$$

where $\pi(x)_\ell$ is the $\ell$th simple action that policy $\pi$ chooses on context $x$. The oracle is supervised as it assumes known features $y_i$ for all simple actions whereas we only observe them for chosen actions. This oracle is the structured generalization of the one considered in contextual bandits [1, 9] and can be implemented by any structured prediction approach such as CRFs [14] or SEARN [8].

Our algorithms choose composite actions by sampling from a distribution, which allows us to use *importance weighting* to construct unbiased estimates for the reward features $y$. If on round $t$, a composite action $A_t$ is chosen with probability $Q_t(A_t)$, we construct the importance weighted feature vector $\hat{y}_t$ with components $\hat{y}_t(a) := y_t(a)\mathbf{1}(a \in A_t)/Q_t(a \in A_t)$, which are unbiased estimators of $y_t(a)$. For a policy $\pi$, we then define empirical estimates of its reward and regret, resp., as

$$\eta_t(\pi, w) := \frac{1}{t} \sum_{i=1}^t \langle w, \hat{y}_i(\pi(x_i)) \rangle \quad \text{and} \quad \widehat{\text{Reg}}_t(\pi, w) := \max_{\pi'} \eta_t(\pi', w) - \eta_t(\pi, w).$$

By construction, $\eta_t(\pi, w^\star)$ is an unbiased estimate of the expected reward $\mathcal{R}(\pi)$, but $\widehat{\text{Reg}}_t(\pi, w^\star)$ is *not* an unbiased estimate of the expected regret $\text{Reg}(\pi)$. We use $\hat{\mathbb{E}}_{x \sim H}[\cdot]$ to denote empirical expectation over contexts appearing in the history of interaction $H$.

---

**Algorithm 1** VCEE (Variance-Constrained Explore-Exploit) Algorithm

---

**Require:** Allowed failure probability $\delta \in (0, 1)$.

1: $Q_0 = 0$, the all-zeros vector. $H_0 = \emptyset$. Define: $\mu_t = \min\left\{1/2K, \sqrt{\ln(16t^2 N/\delta)/(Kt p_{\min})}\right\}$.

2: **for** round $t = 1, \ldots, T$ **do**

3:    Let $\pi_{t-1} = \arg\max_{\pi \in \Pi} \eta_{t-1}(\pi, w^\star)$ and $\tilde{Q}_{t-1} = Q_{t-1} + (1 - \sum_\pi Q_{t-1}(\pi))\mathbf{1}_{\pi_{t-1}}$.

4:    Observe $x_t \in \mathcal{X}$, play $A_t \sim \tilde{Q}_{t-1}^{\mu_{t-1}}(\cdot \mid x_t)$ (see Eq. (3)), and observe $y_t(A_t)$ and $r_t(A_t)$.

5:    Define $q_t(a) = \tilde{Q}_{t-1}^{\mu_{t-1}}(a \in A \mid x_t)$ for each $a$.

6:    Obtain $Q_t$ by solving OP with $H_t = H_{t-1} \cup \{(x_t, y_t(A_t), q_t(A_t)\}$ and $\mu_t$.

7: **end for**

---

**Semi-bandit Optimization Problem (OP)**

With history $H$ and $\mu \geq 0$, define $b_\pi := \frac{\|w^\star\|_1}{\|w^\star\|_2^2} \frac{\widehat{\mathrm{Reg}}_t(\pi)}{\psi \mu p_{\min}}$ and $\psi := 100$. Find $Q \in \Delta_\leq(\Pi)$ such that:

$$\sum_{\pi \in \Pi} Q(\pi) b_\pi \leq 2KL/p_{\min} \tag{4}$$

$$\forall \pi \in \Pi: \quad \hat{\mathbb{E}}_{x \sim H}\left[\sum_{\ell=1}^L \frac{1}{Q^\mu(\pi(x)_\ell \in A \mid x)}\right] \leq \frac{2KL}{p_{\min}} + b_\pi \tag{5}$$

---

Finally, we introduce *projections* and *smoothing* of distributions. For any $\mu \in [0, 1/K]$ and any subdistribution $P \in \Delta_\leq(\Pi)$, the smoothed and projected conditional subdistribution $P^\mu(A \mid x)$ is

$$P^\mu(A \mid x) := (1 - K\mu) \sum_{\pi \in \Pi} P(\pi)\mathbf{1}(\pi(x) = A) + K\mu U_x(A), \tag{3}$$

where $U_x$ is a uniform distribution over a certain subset of valid rankings for context $x$, designed to ensure that the probability of choosing each valid simple action is large. By mixing $U_x$ into our action selection, we limit the variance of reward feature estimates $\hat{y}$. The lower bound on the simple action probabilities under $U_x$ appears in our analysis as $p_{\min}$, which is the largest number satisfying

$$U_x(a \in A) \geq p_{\min}/K$$

for all $x$ and all simple actions $a$ valid for $x$. Note that $p_{\min} = L$ when there are no restrictions on the action space as we can take $U_x$ to be the uniform distribution over all rankings and verify that $U_x(a \in A) = L/K$. In the worst case, $p_{\min} = 1$, since we can always find one valid ranking for each valid simple action and let $U_x$ be the uniform distribution over this set. Such a ranking can be found efficiently by a call to AMO for each simple action $a$, with the dataset of a single point $(x, \mathbf{1}_a \in \mathbb{R}^K, \mathbf{1} \in \mathbb{R}^L)$, where $\mathbf{1}_a(a') = \mathbf{1}(a = a')$.

## 3 Semibandits with known weights

We begin with the setting where the weights $w^\star$ are known, and present an efficient oracle-based algorithm (VCEE, see Algorithm 1) that generalizes the algorithm of Agarwal et al. [1].

The algorithm, before each round $t$, constructs a subdistribution $Q_{t-1} \in \Delta_\leq(\Pi)$, which is used to form the distribution $\tilde{Q}_{t-1}$ by placing the missing mass on the maximizer of empirical reward. The composite action for the context $x_t$ is chosen according to the smoothed distribution $\tilde{Q}_{t-1}^{\mu_{t-1}}$ (see Eq. (3)). The subdistribution $Q_{t-1}$ is any solution to the feasibility problem (OP), which balances exploration and exploitation via the constraints in Eqs. (4) and (5). Eq. (4) ensures that the distribution has low empirical regret. Simultaneously, Eq. (5) ensures that the variance of the reward estimates $\hat{y}$ remains sufficiently small for each policy $\pi$, which helps control the deviation between empirical and expected regret, and implies that $Q_{t-1}$ has low expected regret. For each $\pi$, the variance constraint is based on the empirical regret of $\pi$, guaranteeing sufficient exploration amongst all good policies.

OP can be solved efficiently using AMO and a coordinate descent procedure obtained by modifying the algorithm of Agarwal et al. [1]. While the full algorithm and analysis are deferred to Appendix E, several key differences between VCEE and the algorithm of Agarwal et al. [1] are worth highlighting.

One crucial modification is that the variance constraint in Eq. (5) involves the marginal probabilities of the simple actions rather than the composite actions as would be the most obvious adaptation to our setting. This change, based on using the reward estimates $\hat{y}_t$ for simple actions, leads to substantially lower variance of reward estimates for all policies and, consequently, an improved regret bound. Another important modification is the new mixing distribution $U_x$ and the quantity $p_{\min}$. For structured composite action spaces, uniform exploration over the valid composite actions may not provide sufficient coverage of each simple action and may lead to dependence on the composite action space size, which is exponentially worse than when $U_x$ is used.

The regret guarantee for Algorithm 1 is the following:

**Theorem 1.** *For any* $\delta \in (0,1)$, *with probability at least* $1 - \delta$, VCEE *achieves regret* $\tilde{O}\big(\frac{\|w^\star\|_2^2}{\|w^\star\|_1} L \sqrt{KT \log(N/\delta) / p_{\min}}\big)$. *Moreover,* VCEE *can be efficiently implemented with* $\tilde{O}\big(T^{3/2}\sqrt{K / (p_{\min} \log(N/\delta))}\big)$ *calls to a supervised learning oracle* AMO.

In Table 1, we compare this result to other applicable regret bounds in the most common setting, where $w^\star = \mathbf{1}$ and all rankings are valid ($p_{\min} = L$). VCEE enjoys a $\tilde{O}(\sqrt{KLT \log N})$ regret bound, which is the best bound amongst oracle-based approaches, representing an exponentially better $L$-dependence over the purely bandit feedback variant [1] and a polynomially better $T$-dependence over an $\epsilon$-greedy scheme (see Theorem 3 in Appendix A). This improvement over $\epsilon$-greedy is also verified by our experiments. Additionally, our bound matches that of Kale et al. [12], who consider the harder adversarial setting but give an algorithm that requires an exponentially worse running time, $\Omega(NT)$, and cannot be efficiently implemented with an oracle.

Other results address the non-contextual setting, where the optimal bounds for both stochastic [13] and adversarial [2] semibandits are $\Theta(\sqrt{KLT})$. Thus, our bound may be optimal when $p_{\min} = \Omega(L)$. However, these results apply even without requiring all rankings to be valid, so they improve on our bound by a $\sqrt{L}$ factor when $p_{\min} = 1$. This $\sqrt{L}$ discrepancy may not be fundamental, but it seems unavoidable with some degree of uniform exploration, as in all existing contextual bandit algorithms. A promising avenue to resolve this gap is to extend the work of Neu [18], which gives high-probability bounds in the noncontextual setting without uniform exploration.

To summarize, our regret bound is similar to existing results on combinatorial (semi)bandits but represents a significant improvement over existing computationally efficient approaches.

## 4   Semibandits with unknown weights

We now consider a generalization of the contextual semibandit problem with a new challenge: the weight vector $w^\star$ is unknown. This setting is substantially more difficult than the previous one, as it is no longer clear how to use the semibandit feedback to optimize for the overall reward. Our result shows that the semibandit feedback can still be used effectively, even when the transformation is unknown. Throughout, we assume that the true weight vector $w^\star$ has bounded norm, i.e., $\|w^\star\|_2 \le B$.

One restriction required by our analysis is the ability to play any ranking. Thus, all rankings must be valid in all contexts, which is a natural restriction in domains such as information retrieval and recommendation. The uniform distribution over all rankings is denoted $U$.

We propose an algorithm that explores first and then, adaptively, switches to exploitation. In the exploration phase, we play rankings uniformly at random, with the goal of accumulating enough information to learn the weight vector $w^\star$ for effective policy optimization. Exploration lasts for a variable length of time governed by two parameters $n_\star$ and $\lambda_\star$. The $n_\star$ parameter controls the minimum number of rounds of the exploration phase and is $O(T^{2/3})$, similar to $\epsilon$-greedy style schemes [15]. The adaptivity is implemented by the $\lambda_\star$ parameter, which imposes a lower bound on the eigenvalues of the 2nd-moment matrix of reward features observed during exploration. As a result, we only transition to the exploitation phase after this matrix has suitably large eigenvalues. Since we make no assumptions about the reward features, there is no bound on how many rounds this may take. This is a departure from previous explore-first schemes, and captures the difficulty of learning $w^\star$ when we observe the regression features only after taking an action.

After the exploration phase of $t$ rounds, we perform least-squares regression using the observed reward features and the rewards to learn an estimate $\hat{w}$ of $w^\star$. We use $\hat{w}$ and importance weighted

**Algorithm 2** EELS (Explore-Exploit Least Squares)

---

**Require:** Allowed failure probability $\delta \in (0, 1)$. Assume $\|w^\star\|_2 \le B$.
1: Set $n_\star \leftarrow T^{2/3} (K \ln(N/\delta)/L)^{1/3} \max\{1, (B\sqrt{L})^{-2/3}\}$
2: **for** $t = 1, \dots, n_\star$ **do**
3:     Observe $x_t$, play $A_t \sim U$ ($U$ is uniform over all rankings), observe $y_t(A_t)$ and $r_t(A_t)$.
4: **end for**
5: Let $\hat{V} = \frac{1}{2n_\star K^2} \sum_{t=1}^{n_\star} \sum_{a,b \in \mathcal{A}} \left( y_t(a) - y_t(b) \right)^2 \frac{\mathbf{1}(a, b \in A_t)}{U(a, b \in A_t)}$.
6: $\tilde{V} \leftarrow 2\hat{V} + 3\ln(2/\delta)/(2n_\star)$.
7: Set $\lambda_\star \leftarrow \max \left\{ 6L^2 \ln(4LT/\delta), (T\tilde{V}/B)^{2/3} (L\ln(2/\delta))^{1/3} \right\}$.
8: Set $\Sigma \leftarrow \sum_{t=1}^{n_\star} y_t(A_t) y_t(A_t)^T$.
9: **while** $\lambda_{\min}(\Sigma) \le \lambda_\star$ **do**
10:     $t \leftarrow t + 1$. Observe $x_t$, play $A_t \sim U$, observe $y_t(A_t)$ and $r_t(A_t)$.
11:     Set $\Sigma \leftarrow \Sigma + y_t(A_t) y_t(A_t)^T$.
12: **end while**
13: Estimate weights $\hat{w} \leftarrow \Sigma^{-1} (\sum_{i=1}^t y_i(A_i) r_i(A_i))$ (Least Squares).
14: Optimize policy $\hat{\pi} \leftarrow \operatorname{argmax}_{\pi \in \Pi} \eta_t(\pi, \hat{w})$ using importance weighted features.
15: For every remaining round: observe $x_t$, play $A_t = \hat{\pi}(x_t)$.

---

reward features from the exploration phase to find a policy $\hat{\pi}$ with maximum empirical reward, $\eta_t(\cdot, \hat{w})$. The remaining rounds comprise the exploitation phase, where we play according to $\hat{\pi}$.

The remaining question is how to set $\lambda_\star$, which governs the length of the exploration phase. The ideal setting uses the unknown parameter $V \coloneqq \mathbb{E}_{(x,y) \sim \mathcal{D}} \operatorname{Var}_{a \sim \text{Unif}(\mathcal{A})}[y(a)]$ of the distribution $\mathcal{D}$, where $\text{Unif}(\mathcal{A})$ is the uniform distribution over all simple actions. We form an unbiased estimator $\hat{V}$ of $V$ and derive an upper bound $\tilde{V}$. While the optimal $\lambda_\star$ depends on $V$, the upper bound $\tilde{V}$ suffices.

For this algorithm, we prove the following regret bound.

**Theorem 2.** *For any $\delta \in (0, 1)$ and $T \ge K \ln(N/\delta)/\min\{L, (BL)^2\}$, with probability at least $1 - \delta$, EELS has regret $\tilde{O}\left(T^{2/3} (K \log(N/\delta))^{1/3} \max\{B^{1/3} L^{1/2}, BL^{1/6}\}\right)$. EELS can be implemented efficiently with one call to the optimization oracle.*

The theorem shows that we can achieve sublinear regret without dependence on the composite action space size even when the weights are unknown. The only applicable alternatives from the literature are displayed in Table 1, specialized to $B = \Theta(\sqrt{L})$. First, oracle-based contextual bandits [1] achieve a better $T$-dependence, but both the regret and the number of oracle calls grow exponentially with $L$. Second, the deviation bound of Swaminathan et al. [22], which exploits the reward structure but not the semibandit feedback, leads to an algorithm with regret that is polynomially worse in its dependence on $L$ and $B$ (see Appendix B). This observation is consistent with non-contextual results, which show that the value of semibandit information is only in $L$ factors [2].

Of course EELS has a sub-optimal dependence on $T$, although this is the best we are aware of for a computationally efficient algorithm in this setting. It is an interesting open question to achieve $\operatorname{poly}(K, L)\sqrt{T \log N}$ regret with unknown weights.

## 5 Proof sketches

We next sketch the arguments for our theorems. Full proofs are deferred to the appendices.

**Proof of Theorem 1**: The result generalizes Agarwal et. al [1], and the proof structure is similar. For the regret bound, we use Eq. (5) to control the deviation of the empirical reward estimates which make up the empirical regret $\widehat{\text{Reg}}_t$. A careful inductive argument leads to the following bounds:

$$\text{Reg}(\pi) \le 2\widehat{\text{Reg}}_t(\pi) + c_0 \frac{\|w^\star\|_2^2}{\|w^\star\|_1} KL\mu_t \quad \text{and} \quad \widehat{\text{Reg}}_t(\pi) \le 2\text{Reg}(\pi) + c_0 \frac{\|w^\star\|_2^2}{\|w^\star\|_1} KL\mu_t.$$

Here $c_0$ is a universal constant and $\mu_t$ is defined in the pseudocode. Eq. (4) guarantees low empirical regret when playing according to $\tilde{Q}_t^{\mu_t}$, and the above inequalities also ensure small population regret.

The cumulative regret is bounded by $\frac{\|w^\star\|_2^2}{\|w^\star\|_1} KL \sum_{t=1}^{T} \mu_t$, which grows at the rate given in Theorem 1. The number of oracle calls is bounded by the analysis of the number of iterations of coordinate descent used to solve OP, via a potential argument similar to Agarwal et al. [1].

**Proof of Theorem 2**: We analyze the exploration and exploitation phases individually, and then optimize $n_\star$ and $\lambda_\star$ to balance these terms. For the exploration phase, the expected per-round regret can be bounded by either $\|w^\star\|_2 \sqrt{KV}$ or $\|w^\star\|_2 \sqrt{L}$, but the number of rounds depends on the minimum eigenvalue $\lambda_{\min}(\Sigma)$, with $\Sigma$ defined in Steps 8 and 11. However, the expected per-round 2nd-moment matrix, $\mathbb{E}_{(x,y)\sim\mathcal{D}, A\sim U}[y(A)y(A)^T]$, has all eigenvalues at least $V$. Thus, after $t$ rounds, we expect $\lambda_{\min}(\Sigma) \geq tV$, so exploration lasts about $\lambda_\star/V$ rounds, yielding roughly

$$\text{Exploration Regret} \leq \frac{\lambda_\star}{V} \cdot \|w^\star\|_2 \min\{\sqrt{KV}, \sqrt{L}\}.$$

Now our choice of $\lambda_\star$ produces a benign dependence on $V$ and yields a $T^{2/3}$ bound.

For the exploitation phase, we bound the error between the empirical reward estimates $\eta_t(\pi, \hat{w})$ and the true reward $\mathcal{R}(\pi)$. Since we know $\lambda_{\min}(\Sigma) \geq \lambda_\star$ in this phase, we obtain

$$\text{Exploitation Regret} \leq T\|w^\star\|_2 \sqrt{\frac{K \log N}{n_\star}} + T\sqrt{\frac{L}{\lambda_\star}} \min\{\sqrt{KV}, \sqrt{L}\}.$$

The first term captures the error from using the importance-weighted $\hat{y}$ vector, while the second uses a bound on the error $\|\hat{w} - w^\star\|_2$ from the analysis of linear regression (assuming $\lambda_{\min}(\Sigma) \geq \lambda_\star$).

This high-level argument ignores several important details. First, we must show that using $\tilde{V}$ instead of the optimal choice $V$ in the setting of $\lambda_\star$ does not affect the regret. Secondly, since the termination condition for the exploration phase depends on the random variable $\Sigma$, we must derive a high-probability bound on the number of exploration rounds to control the regret. Obtaining this bound requires a careful application of the matrix Bernstein inequality to certify that $\Sigma$ has large eigenvalues.

# 6 Experimental Results

Our experiments compare VCEE with existing alternatives. As VCEE generalizes the algorithm of Agarwal et al. [1], our experiments also provide insights into oracle-based contextual bandit approaches and this is the first detailed empirical study of such algorithms. The weight vector $w^\star$ in our datasets was known, so we do not evaluate EELS. This section contains a high-level description of our experimental setup, with details on our implementation, baseline algorithms, and policy classes deferred to Appendix C. Software is available at http://github.com/akshaykr/oracle_cb.

**Data:** We used two large-scale learning-to-rank datasets: MSLR [17] and all folds of the Yahoo! Learning-to-Rank dataset [5]. Both datasets have over 30k unique queries each with a varying number of documents that are annotated with a relevance in $\{0, \ldots, 4\}$. Each query-document pair has a feature vector ($d = 136$ for MSLR and $d = 415$ for Yahoo!) that we use to define our policy class. For MSLR, we choose $K = 10$ documents per query and set $L = 3$, while for Yahoo!, we set $K = 6$ and $L = 2$. The goal is to maximize the sum of relevances of shown documents ($w^\star = \mathbf{1}$) and the individual relevances are the semibandit feedback. All algorithms make a single pass over the queries.

**Algorithms:** We compare VCEE, implemented with an epoch schedule for solving OP after $2^{i/2}$ rounds (justified by Agarwal et al. [1]), with two baselines. First is the $\epsilon$-GREEDY approach [15], with a constant but tuned $\epsilon$. This algorithm explores uniformly with probability $\epsilon$ and follows the empirically best policy otherwise. The empirically best policy is updated with the same $2^{i/2}$ schedule.

We also compare against a semibandit version of LINUCB [19]. This algorithm models the semibandit feedback as linearly related to the query-document features and learns this relationship, while selecting composite actions using an upper-confidence bound strategy. Specifically, the algorithm maintains a weight vector $\theta_t \in \mathbb{R}^d$ formed by solving a ridge regression problem with the semibandit feedback $y_t(a_{t,\ell})$ as regression targets. At round $t$, the algorithm uses document features $\{x_a\}_{a \in \mathcal{A}}$ and chooses the $L$ documents with highest $x_a^T \theta_t + \alpha x_a^T \Sigma_t^{-1} x_a$ value. Here, $\Sigma_t$ is the feature 2nd-moment matrix and $\alpha$ is a tuning parameter. For computational reasons, we only update $\Sigma_t$ and $\theta_t$ every 100 rounds.

**Oracle implementation:** LINUCB only works with a linear policy class. VCEE and $\epsilon$-GREEDY work with arbitrary classes. Here, we consider three: linear functions and depth-2 and depth-5

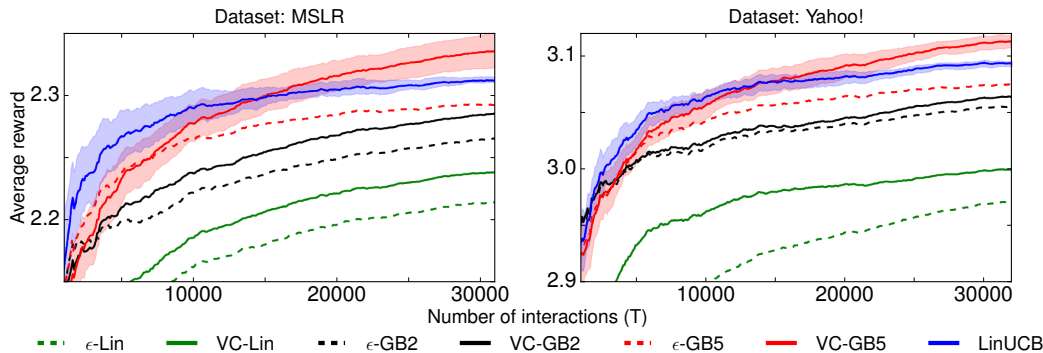

Figure 1: Average reward as a function of number of interactions $T$ for VCEE, $\epsilon$-GREEDY, and LINUCB on MSLR (left) and Yahoo (right) learning-to-rank datasets.

gradient boosted regression trees (abbreviated Lin, GB2 and GB5). Both GB classes use 50 trees. Precise details of how we instantiate the supervised learning oracle can be found in Appendix C.

**Parameter tuning:** Each algorithm has a parameter governing the explore-exploit tradeoff. For VCEE, we set $\mu_t = c\sqrt{1/KLT}$ and tune $c$, in $\epsilon$-GREEDY we tune $\epsilon$, and in LINUCB we tune $\alpha$. We ran each algorithm for 10 repetitions, for each of ten logarithmically spaced parameter values.

**Results:** In Figure 1, we plot the average reward (cumulative reward up to round $t$ divided by $t$) on both datasets. For each $t$, we use the parameter that achieves the best average reward across the 10 repetitions at that $t$. Thus for each $t$, we are showing the performance of each algorithm tuned to maximize reward over $t$ rounds. We found VCEE was fairly stable to parameter tuning, so for VC-GB5 we just use one parameter value ($c = 0.008$) for all $t$ on both datasets. We show confidence bands at twice the standard error for just LINUCB and VC-GB5 to simplify the plot.

Qualitatively, both datasets reveal similar phenomena. First, when using the same policy class, VCEE consistently outperforms $\epsilon$-GREEDY. This agrees with our theory, as VCEE achieves $\sqrt{T}$-type regret, while a tuned $\epsilon$-GREEDY achieves at best a $T^{2/3}$ rate.

Secondly, if we use a rich policy class, VCEE can significantly improve on LINUCB, the empirical state-of-the-art, and one of few practical alternatives to $\epsilon$-GREEDY. Of course, since $\epsilon$-GREEDY does not outperform LINUCB, the tailored exploration of VCEE is critical. Thus, the combination of these two properties is key to improved performance on these datasets. VCEE is the only contextual semibandit algorithm we are aware of that performs adaptive exploration *and* is agnostic to the policy representation. Note that LINUCB is quite effective and outperforms VCEE with a linear class. One possible explanation for this behavior is that LINUCB, by directly modeling the reward, searches the policy space more effectively than VCEE, which uses an approximate oracle implementation.

# 7 Discussion

This paper develops oracle-based algorithms for contextual semibandits both with known and unknown weights. In both cases, our algorithms achieve the best known regret bounds for computationally efficient procedures. Our empirical evaluation of VCEE, clearly demonstrates the advantage of sophisticated oracle-based approaches over both parametric approaches and naive exploration. To our knowledge this is the first detailed empirical evaluation of oracle-based contextual bandit or semibandit learning. We close with some promising directions for future work:

1. With known weights, can we obtain $\tilde{O}(\sqrt{KLT \log N})$ regret even with structured action spaces? This may require a new contextual bandit algorithm that does not use uniform smoothing.

2. With unknown weights, can we achieve a $\sqrt{T}$ dependence while exploiting semibandit feedback?

## Acknowledgements

This work was carried out while AK was at Microsoft Research.

## Footnotes

[1]Throughout the paper, the $\tilde{O}(\cdot)$ notation suppressed factors polylogarithmic in $K$, $L$, $T$ and $\log N$. We analyze finite policy classes, but our work extends to infinite classes by standard discretization arguments.

[2]Kale et al. [12] consider the favorable setting where our bounds match, when uniform exploration is valid.

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
