[Supplementary Material]

**Algorithm 3** $\epsilon$-Greedy for Contextual Semibandits with Known Weights

---

**Require:** Allowed failure probability $\delta \in (0, 1)$.
    Set $n = T^{2/3}(K \ln(2N/\delta)/L)^{1/3}$.
    Let $U$ be the uniform distribution over all rankings.
    For $t = 1, \dots, n$, observe $x_t$, play $A_t \sim U$, observe $y_t(A_t)$ and $r_t(A_t)$.
    Optimize policy $\hat{\pi} \leftarrow \operatorname{argmax}_{\pi \in \Pi} \eta_n(\pi, w^\star)$ using importance-weighted features.
    For every remaining round: observe $x_t$, play $A_t = \hat{\pi}(x_t)$.

---

## A    Analysis of $\epsilon$-Greedy with Known Weights

We analyze the $\epsilon$-greedy algorithm (Algorithm 3) in the known-weights setting when all rankings are valid, i.e., $p_{\min} = L$. This algorithm is different from the one we use in our experiments in that it is an explore-first variant, exploring for the first several rounds and then exploiting for the remainder. In our experiments, we use a variant where at each round we explore with probability $\epsilon$ and exploit with probability $(1 - \epsilon)$. This latter version also has the same regret bound, via an argument similar to that of Langford and Zhang [15].

**Theorem 3.** *For any $\delta \in (0, 1)$, when $T \geq K \ln(N/\delta)/L$, with probability at least $1 - \delta$, the regret of Algorithm 3 is at most $\tilde{O}(\|w^\star\|_2 T^{2/3}(K \log(N/\delta))^{1/3} L^{1/6})$.*

*Proof.* The proof relies on the uniform deviation bound similar to Lemma 20, which we use for the analysis of EELS. We first prove that for any $\delta \in (0, 1)$, with probability at least $1 - \delta$, for all policies $\pi$, we have

$$|\eta_n(\pi, w^\star) - \mathcal{R}(\pi)| \leq \|w^\star\|_2 \left( \sqrt{\frac{2K \ln(2N/\delta)}{n}} + \frac{2K}{3\sqrt{L}} \frac{\ln(2N/\delta)}{n} \right). \qquad (6)$$

This deviation bound is a consequence of Bernstein's inequality. The quantity on the left-hand side is the average of $n$ terms

$$\hat{y}_i(\pi(x_i))^T w^\star - \mathbb{E}_{x,y}[y(\pi(x))]^T w^\star,$$

all with expectation zero, because $\hat{y}$ is unbiased. The range of each term is bounded by the Cauchy-Schwarz inequality as

$$\|w^\star\|_2 \|\hat{y}_i(\pi(x_i)) - \mathbb{E}_{x,y}[y(\pi(x))]\|_2 \leq \|w^\star\|_2 K/\sqrt{L},$$

because under uniform exploration the coordinates of $\hat{y}_i(\pi(x_i))$ are bounded in $[0, K/L]$ while the coordinates of $y(\pi(x))$ are in $[0, 1]$ and these are $L$-dimensional vectors. The variance is bounded by the second moment, which we bound as follows:

$$\mathbb{E}_{x,y,A}\left[(\hat{y}(\pi(x))^T w^\star)^2\right] \leq \|w^\star\|_2^2 \mathbb{E}_{x,y,A}\left[\sum_{l=1}^{L} \hat{y}(\pi(x_l))^2\right]$$

$$\leq \|w^\star\|_2^2 \mathbb{E}_{x,y,A}\left[\sum_{l=1}^{L} \frac{K^2}{L^2} \mathbf{1}(\pi(x)_l \in A)\right] = \|w^\star\|^2 K,$$

since $\mathbb{E}_{x,y,A}[\mathbf{1}(\pi(x)_l \in A)] = L/K$ under uniform exploration. Plugging these bounds into Bernstein's inequality gives the deviation bound of Eq. (6).

Now we can prove the theorem. Eq. (6) ensures that after collecting $n$ samples, the expected reward of the empirical reward maximizer $\hat{\pi}$ is close to $\max_\pi \mathcal{R}(\pi)$, the best achievable reward. The difference between these two is at most twice the right-hand side of the deviation bound. If we perform uniform exploration for $n$ rounds, we are ensured that with probability at least $1 - \delta$ the regret is at most

$$\text{Regret} \leq n\|w^\star\|_2 \sqrt{L} + 2(T - n)\|w^\star\|_2 \left( \sqrt{\frac{2K \ln(2N/\delta)}{n}} + \frac{2K}{3\sqrt{L}} \frac{\ln(2N/\delta)}{n} \right)$$

$$\leq n\|w^\star\|_2 \sqrt{L} + 3T\|w^\star\|_2 \left( \sqrt{\frac{K \ln(2N/\delta)}{n}} + \frac{K}{\sqrt{L}} \frac{\ln(2N/\delta)}{n} \right).$$

For our setting of $n = T^{2/3}(K \ln(2N/\delta)/L)^{1/3}$, the bound is

$$4\|w^\star\|_2 T^{2/3}(K \ln(2N/\delta))^{1/3} L^{1/6} + 3\|w^\star\|_2 T^{1/3}(K \ln(2N/\delta))^{2/3} L^{-1/6}.$$

Under the assumption on $T$, the second term is lower order, which proves the result. $\square$

# B Comparisons for EELS

In this section we do a detailed comparison of our Theorem 2 to the paper of Swaminathan et al. [22], which is the most directly applicable result. We use notation consistent with our paper.

Swaminathan et al. [22] focus on off-policy evaluation in a more challenging setting where no semibandit feedback is provided. Specifically, in their setting, in each round, the learner observes a context $x \in \mathcal{X}$, chooses a composite action $A$ (as we do here) and receives reward $r(A) \in [-1, 1]$. They assume that the reward decomposes linearly across the action-position pairs as

$$\mathbb{E}[r(A)|x, A] = \sum_{\ell=1}^{L} \phi_x(a_\ell, \ell).$$

With this assumption, and when exploration is done uniformly, they provide off-policy reward estimation bounds of the form

$$|\eta_n(\pi) - \mathcal{R}(\pi)| \leq O\left(\sqrt{\frac{KL\ln(1/\delta)}{n}}\right).$$

This bound holds for any policy $\pi : \mathcal{X} \to \mathcal{A}^L$ with probability at least $1 - \delta$ for any $\delta \in (0, 1)$. (See Theorem 3 and the following discussion in Swaminathan et al. [22].) Note that this assumption generalizes our unknown weights setting, since we can always define $\phi_x(a, j) = w_j^\star y(a)$.

To do an appropriate comparison, we first need to adjust the scaling of the rewards. While Swaminathan et al. [22] assume that rewards are bounded in $[-1, 1]$, we only assume bounded $y$'s and bounded noise. Consequently, we need to adjust their bound to incorporate this scaling. If the rewards are scaled to lie in $[-R, R]$, their bound becomes

$$|\eta_n(\pi) - \mathcal{R}(\pi)| \leq O\left(R\sqrt{\frac{KL\ln(1/\delta)}{n}}\right).$$

This deviation bound can be turned into a low-regret algorithm by exploring for the first $n$ rounds, finding an empirically best policy, and using that policy for the remaining $T - n$ rounds. Optimizing the bound in $n$ leads to a $T^{2/3}$-style regret bound:

**Fact 4.** *The approach of Swaminathan et al. [22] with rewards in $[-R, R]$ leads to an algorithm with regret bound*

$$O\left(RT^{2/3}(KL\log N)^{1/3}\right).$$

This algorithm can be applied as is to our setting, so it is worth comparing it to EELS. According to Theorem 2, EELS has a regret bound

$$O\left(T^{2/3}(K\log N)^{1/3}\max\{B^{1/3}L^{1/2}, BL^{1/6}\}\right).$$

The dependence on $T$, $K$, and $\log N$ match between the two algorithms, so we are left with $L$ and the scale factors $B, R$. This comparison is somewhat subtle and we use two different arguments. The first finds a conservative value for $R$ in Fact 4 in terms of $B$ and $L$. This is the regret bound one would obtain by using the approach of Swaminathan et al. [22] in our precise setting, ignoring the semibandit feedback, but with known weight-vector bound $B$. The second comparison finds a conservative value of $B$ in terms of $R$ and $L$.

For the first comparison, recall that our setting makes no assumptions on the scale of the reward, except that the noise $\xi$ is bounded in $[-1, 1]$, so our setting never admits $R < 1$. If we begin with a setting of $B$, we need to conservatively set $R = \max\{B\sqrt{L}, 1\}$, which gives the dependence

EELS: $\max\{B^{1/3}L^{1/2}, BL^{1/6}\}$

Swaminathan et al. [22]: $\max\{BL^{5/6}, L^{1/3}\}$.

The EELS bound is never worse than the bound in Fact 4 according to this comparison. At $B = \Theta(L^{-1/2})$, the two bounds are of the same order, which is $\Theta(L^{1/3})$. For $B = O(L^{-1/2})$, the EELS bound is at most $L^{1/3}$, while for $B = \Omega(L^{-1/2})$ the first term in the EELS bound is at most the first term in the Swaminathan et al. [22] bound. In both cases, the EELS bound is superior. Finally when $B = \Omega(\sqrt{L})$, the second term dominates our bound, so EELS demonstrates an $L^{2/3}$ improvement.

For the second comparison, since our setting has the noise bounded in $[-1, 1]$, assume that $R \geq 1$ and that the total reward is scaled in $[-R, R]$ as in Fact 4. If we want to allow any $y(A) \in [0, 1]^L$, the tightest setting of $R$ is between $\|w^\star\|_1 / 2$ and $\|w^\star\|_1$ (depending on the structure of the positive and negative coordinates of $w^\star$). For simplicity, assume $R$ is a bound on $\|w^\star\|_1$. Since the EELS bound depends on $B$, a bound on the Euclidean norm of $w^\star$, we use $\|w^\star\|_2 \leq \|w^\star\|_1 \leq \sqrt{L}\|w^\star\|_2$ to obtain a conservative setting of $B = R$. This gives the dependence

$$\text{EELS: } \max\{R^{1/3}L^{1/2}, RL^{1/6}\}$$

$$\text{Swaminathan et al. [22]: } RL^{1/3}$$

Since $R \geq 1$, the EELS bound is superior whenever $R \geq L^{1/4}$. Moreover, if $R = \Omega(\sqrt{L})$, i.e., at least $\sqrt{L}$ positions are relevant, the second term dominates our bound, and we improve by a factor of $L^{1/6}$. The EELS bound is inferior when $R \leq L^{1/4}$, which corresponds to a high-sparsity case since $R$ is also a bound on $\|w^\star\|_1$ in this comparison.

## C   Implementation Details

### C.1   Implementation of VCEE

VCEE is implemented as stated in Algorithm 1 with some modifications, primarily to account for an imperfect oracle. OP is solved using the coordinate descent procedure described in Appendix E.

We set $\psi = 1$ in our implementation and ignore the log factor in $\mu_t$. Instead, since $p_{\min} = L$, we use $\mu_t = c\sqrt{1/KLT}$ and tune $c$, which can compensate for the absence of the $\log(t^2 N/\delta)$ factor. This additionally means that we ignore the failure probability parameter $\delta$. Otherwise, all other parameters and constants are set as described in Algorithm 1 and OP.

As mentioned in Section 6, we implement AMO via a reduction to squared loss regression. There are many possibilities for this reduction. In our case, we specify a squared loss regression problem via a dataset $D = \{x_i, A_i, y_i, \gamma_i\}_{i=1}^n$ where $x_i \in \mathcal{X}$, $A_i$ is any list of actions, $y_i \in \mathbb{R}^K$ assigns a value to each action, and $\gamma_i \in \mathbb{R}^K$ assigns an importance weight to each action. Since in our experiments $w^\star = \mathbf{1}$, we do not need to pass along the vectors $v_i \in \mathbb{R}^L$ described in Eq. (2).

Given such a dataset $D$, we minimize a *weighted squared loss* objective over a regression class $\mathcal{F}$,

$$\hat{f} = \underset{f \in \mathcal{F}}{\arg\min} \sum_{i=1}^n \sum_{a \in A_i} \gamma_i(a)(f(\phi(x_i, a)) - y_i(a))^2, \tag{7}$$

where $\phi(x, a)$ is a feature vector associated with the given query-document pair. Note that we only include terms corresponding to simple actions in $A_i$ for each $i$. This regression function is associated with the greedy policy that chooses the best valid ranking according to the sum of rewards of individual actions as predicted by $\hat{f}$ on the current context.

We access this oracle with two different kinds of datasets. When we access AMO to find the empirically best policy, we only use the history of the interaction. In this case, we only regress onto the chosen actions in the history and we let $\gamma_i$ be their importance weights. More formally, suppose that at round $t$, we observe context $x_t$, choose composite action $A_t \sim q_t$ and receive feedback $\{y_t(a_{t,\ell})\}_{\ell=1}^L$. We create a single example $(x_t, A_t, z_t, \gamma_t)$ where $x_t$ is the context, $A_t$ is the chosen composite action, $z_t$ has $z_t(a) = \mathbf{1}(a \in A_t)y_t(a)$ and $\gamma_t(a) = 1/q_t(a \in A_t)$. Observe that when this sample is passed into Eq. (7), it leads to a different objective than if we regressed directly onto the importance-weighted reward features $\hat{y}_t$.

We also create datasets to verify the variance constraint within OP. For this, we use the AMO in a more direct way by setting $A_t$ to be a list of all $K$ actions, letting $y_t$ be the importance weighted vector, and $\gamma_t = \mathbf{1}$.

We use this particular implementation because leaving the importance weights inside the square loss term introduces additional variance, which we would like to avoid.

The imperfect oracle introduces one issue that needs to be corrected. Since the oracle is not guaranteed to find the maximizing policy on every dataset, in the $t$th round of the algorithm, we may encounter a policy $\pi$ that has $\widehat{\mathrm{Reg}}_t(\pi) < 0$, which can cause the coordinate descent procedure to loop indefinitely. Of course, if we ever find a policy $\pi$ with $\widehat{\mathrm{Reg}}_t(\pi) < 0$, it means that we have found a better policy, so we simply switch the leader. We found that with this intuitive change, the coordinate descent procedure always terminates in a few iterations.

## C.2   Implementation of $\epsilon$-GREEDY

Recall that we run a variant of $\epsilon$-GREEDY where at each round we explore with probability $\epsilon$ and exploit with probability $(1 - \epsilon)$, which is slightly different from the explore-first algorithm analyzed in Appendix A.

For $\epsilon$-GREEDY, we also use the oracle defined in Eq. (7). This algorithm only accesses the oracle to find the empirically best policy, and we do this in the same way as VCEE does, i.e., we only regress onto actions that were actually selected with importance weights encoded via $\gamma_i$s. We use all of the data, including the data from exploitation rounds, with importance weighting.

## C.3   Implementation of LINUCB

The semibandit version of LINUCB uses ridge regression to predict the semibandit feedback given query-document features $\phi(x, a)$. If the feature vectors are in $d$ dimensions, we start with $\Sigma_1 = I_d$ and $\theta_1 = 0$, the all zeros vector. At round $t$, we receive the query-document feature vectors $\{\phi(x_t, a)\}_{a \in \mathcal{A}}$ for query $x_t$ and we choose

$$A_t = \underset{A}{\mathrm{argmax}} \left\{ \sum_{a \in A} \theta_t^T \phi(x_t, a) + \alpha \phi(x_t, a)^T \Sigma_t^{-1} \phi(x_t, a) \right\}.$$

Since in our experiment we know that $w^\star = \mathbf{1}$ and all rankings are valid, the order of the documents is irrelevant and the best ranking consists of the top $L$ simple actions with the largest values of the above "regularized score". Here $\alpha$ is a parameter of the algorithm that we tune.

After selecting a ranking, we collect the semibandit feedback $\{y_t(a_{t,\ell})\}_{\ell=1}^L$. The standard implementation would perform the update

$$\Sigma_{t+1} \leftarrow \Sigma_t + \sum_{\ell=1}^L \phi(x_t, a_{t,\ell}) \phi(x_t, a_{t,\ell})^T, \qquad \theta_{t+1} \leftarrow \Sigma_{t+1}^{-1} \left( \sum_{i=1}^t \sum_{\ell=1}^L \phi(x_i, a_{i,\ell}) y_i(a_{i,\ell}) \right),$$

which is the standard online ridge regression update. For computational reasons, we only update every 100 iterations, using all of the data. Thus, if $\mathrm{mod}(t, 100) \neq 0$, we set $\Sigma_{t+1} \leftarrow \Sigma_t$ and $\theta_{t+1} \leftarrow \theta_t$. If $\mathrm{mod}(t, 100) = 0$, we set

$$\Sigma_{t+1} \leftarrow I + \sum_{i=1}^t \sum_{\ell=1}^L \phi(x_i, a_{i,\ell}) \phi(x_i, a_{i,\ell})^T, \qquad \theta_{t+1} \leftarrow \Sigma_{t+1}^{-1} \left( \sum_{i=1}^t \sum_{\ell=1}^L \phi(x_i, a_{i,\ell}) y_i(a_{i,\ell}) \right).$$

## C.4   Policy Classes

As AMO for both VCEE and $\epsilon$-GREEDY, we use the default implementations of regression with various function classes in `scikit-learn` version 0.17. We instantiate `scikit-learn` model objects and use the `fit()` and `predict()` routines. The model objects we use are

1. `sklearn.linear_model.LinearRegression()`

2. `sklearn.ensemble.GradientBoostingRegressor(n_estimators=50,max_depth=2)`

3. `sklearn.ensemble.GradientBoostingRegressor(n_estimators=50,max_depth=5)`

All three objects accommodate weighted least-squares objectives as required by Eq. (7).

# D  Proof of Regret Bound in Theorem 1

The proof hinges on two uniform deviation bounds, and then a careful inductive analysis of the regret using the OP. We only need our two deviation bounds to hold for the rounds $t$ in which $\mu_t = \sqrt{\ln(16t^2 N/\delta)/(Ktp_{\min})}$. Let $d_t := \ln(16t^2 N/\delta)$. These rounds then start at

$$t_0 := \min\left\{t: \sqrt{\frac{d_t}{Ktp_{\min}}} \leq \frac{1}{2K}\right\} = \min\left\{t: \frac{d_t}{t} \leq \frac{p_{\min}}{4K}\right\}.$$

Note that $t_0 \geq 4$ since $d_t \geq 1$ and $K \geq p_{\min}$. From the definition of $t_0$, we have for all $t \geq t_0$:

$$\mu_t \geq \sqrt{d_t/(Ktp_{\min})}, \quad t \geq 4Kd_t/p_{\min}. \tag{8}$$

The first deviation bound shows that the variance estimates used in Eq. (5) are suitable estimators for the true variance of the distribution. To state this deviation bound, we need some definitions:

$$V(P, \pi, \mu) := \mathbb{E}_{x \sim \mathcal{D}_x}\left[\sum_{\ell=1}^{L} \frac{1}{P^\mu(\pi(x)_\ell \mid x)}\right], \quad \hat{V}_t(P, \pi, \mu) := \hat{\mathbb{E}}_{x \sim H_t}\left[\sum_{\ell=1}^{L} \frac{1}{P^\mu(\pi(x)_\ell \mid x)}\right]. \tag{9}$$

In these definitions and throughout this appendix we use the shorthand $P(a \mid x)$ to mean $P(a \in A \mid x)$ for any projected subdistribution $P(A \mid x)$. If $P$ is a distribution, we have $\sum_{a \in \mathcal{A}} P(a \mid x) = L$. For a subdistribution, this sum can be smaller, so $\sum_{a \in \mathcal{A}} P(a \mid x) \leq L$ for all subdistributions. The deviation bound is in the following theorem:

**Theorem 5.** *Let $\delta \in (0, 1)$. Then with probability at least $1 - \delta/8$, for all $t \geq t_0$, all distributions $P$ over $\Pi$, and all $\pi \in \Pi$, we have*

$$V(P, \pi, \mu_t) \leq 6.4\hat{V}_t(P, \pi, \mu_t) + 81.3\frac{KL}{p_{\min}}. \tag{10}$$

*Proof.* The proof of this theorem is similar to a related result of Agarwal et al. [1] (See their Lemma 10). We first use Freedman's inequality (Lemma 23) to argue that for a fixed $P, \pi, \mu$, and $t$, the empirical version of the variance is close to the true variance. We then use a discretization of the set of all distributions and take a union bound to extend this deviation inequality to all $P, \pi, \mu, t$.

To start, we have:

**Lemma 6.** *For fixed $P, \pi, \mu, t$ and for any $\lambda \in \left[0, \frac{\mu p_{\min}}{L}\right]$, with probability at least $1 - \delta$:*

$$V(P, \pi, \mu) - \hat{V}_t(P, \pi, \mu) \leq \frac{(e-2)\lambda L}{\mu p_{\min}} V(P, \pi, \mu) + \frac{\ln(1/\delta)}{t\lambda}$$

*Proof.* Let:

$$Z_i := \sum_{\ell=1}^{L} \frac{1}{P^\mu(\pi(x_i)_\ell | x_i)} - \mathbb{E}_{x \sim \mathcal{D}_x} \sum_{\ell=1}^{L} \frac{1}{P^\mu(\pi(x)_\ell | x)},$$

and notice that $\frac{1}{t}\sum_{i=1}^{t} Z_i = \hat{V}_t(P, \pi, \mu) - V(P, \pi, \mu)$. Clearly, $\mathbb{E}Z_i = 0$ for all $i$ and $\max_i |Z_i| \leq L/\mu p_{\min}$ since when we smooth by $\mu$, each simple action that $\pi$ could choose must appear with probability at least $\mu p_{\min}$. By the Cauchy-Schwarz and Holder inequalities, the conditional variance is:

$$\mathbb{E}_{x \sim \mathcal{D}_x} Z_i^2 \leq \mathbb{E}_{x \sim \mathcal{D}_x}\left(\sum_{\ell=1}^{L} \frac{1}{P^\mu(\pi(x)_\ell | x)}\right)^2 \leq L\mathbb{E}_{x \sim \mathcal{D}_x} \sum_{\ell=1}^{L} \frac{1}{P^\mu(\pi(x)_\ell | x)^2}$$

$$\leq \frac{L}{\mu p_{\min}} \mathbb{E}_{x \sim \mathcal{D}_x} \sum_{\ell=1}^{L} \frac{1}{P^\mu(\pi(x)_\ell | x)} = \frac{L}{\mu p_{\min}} V(P, \pi, \mu).$$

The lemma now follows by Freedman's inequality. □

To prove the variance deviation bound of Theorem 5, we next use a discretization lemma from [9] (their Lemma 16) which immediately implies that for any $P$, there exists a distribution $P'$ supported on at most $N_t$ policies such that for $c_t > 0$, if $N_t \geq \frac{6}{\gamma_t^2 \mu_t p_{\min}}$:

$$V(P,\pi,\mu) - V(P',\pi,\mu_t) + c_t\left(\hat{V}_t(P',\pi,\mu_t) - \hat{V}_t(P,\pi,\mu_t)\right) \leq \gamma_t(V(P,\pi,\mu_t) + c_t\hat{V}_t(P,\pi,\mu_t))$$

This is exactly the second conclusion of their Lemma 16 except we use $c_t$ instead of their $(1+\lambda)$ (we will set $c_t > 1$). The other difference is the inclusion of $p_{\min}$ in the lower bound on $N_t$, which is based on a straightforward modification to their proof.

We set $\gamma_t = \sqrt{\frac{1-K\mu_t}{N_t \mu_t p_{\min}}} + 3\frac{1-K\mu_t}{N_t \mu_t p_{\min}}$, $c_t = \frac{1}{1 - \frac{(e-2)L\lambda_t}{\mu_t p_{\min}}}$, $N_t = \lceil\frac{12(1-K\mu_t)}{\mu_t p_{\min}}\rceil$ and $\lambda_t = 0.66\mu_t p_{\min}/L$.

The choice of $c_t$ is motivated by Lemma 6, which can be rearranged to (for a distribution $P'$)

$$V(P',\pi,\mu_t) - \frac{1}{1 - \frac{(e-2)L\lambda_t}{\mu_t p_{\min}}}\hat{V}_t(P',\pi,\mu_t) \leq \frac{1}{1 - \frac{(e-2)L\lambda_t}{\mu_t p_{\min}}}\frac{\ln(1/\delta)}{t\lambda_t}$$

$$\Leftrightarrow V(P',\pi,\mu_t) - c_t\hat{V}_t(P',\pi,\mu_t) \leq c_t\frac{\ln(1/\delta)}{t\lambda_t}.$$

To take a union over all $t \in \mathbb{N}$, $N_t$-point distributions $P$ over $\Pi$, and all $\pi \in \Pi$, we set $\delta_t = \delta(\frac{1}{2t^2 N^{N_t+1}})$ in the $t$th iteration. This inequality becomes

$$V(P',\pi,\mu_t) - c_t\hat{V}_t(P',\pi,\mu_t) \leq c_t\frac{\ln(2N^{N_t+1}t^2/\delta)}{t\lambda_t}.$$

The choice of $c_t$ and $\lambda_t$ leads to a bound $c_t = \frac{1}{1-0.66(e-2)} \leq 1.91$.

We also use the values of $N_t$ and $\gamma_t$ to bound

$$\gamma_t = \sqrt{\frac{1-K\mu_t}{N_t\mu_t p_{\min}}} + 3\frac{1-K\mu_t}{N_t\mu_t p_{\min}} \leq \sqrt{\frac{1}{12}} + \frac{1}{4}.$$

Rearranging the discretization claim gives

$$V(P,\pi,\mu_t) \leq \frac{c_t(1+\gamma_t)}{(1-\gamma_t)}\hat{V}_t(P,\pi,\mu_t) + \frac{1}{(1-\gamma_t)}\left(V(P',\pi,\mu_t) - c_t\hat{V}_t(P',\pi,\mu_t)\right)$$

$$\leq 6.4\hat{V}_t(P,\pi,\mu_t) + \frac{c_t}{(1-\gamma_t)}\frac{\ln(2N^{N_t+1}t^2/\delta)}{t\lambda_t}.$$

Using the bounds on $c_t, \gamma_t$ and the settings of $N_t$ and $\lambda_t$, this last term is at most

$$\frac{c_t}{(1-\gamma_t)}\left(\frac{L\ln(2N^2t^2/\delta)}{t\mu_t p_{\min}} + \frac{LN_t\ln(N)}{t\mu_t p_{\min}}\right) \leq \frac{6.3L\ln(16N^2t^2/\delta)}{\mu_t t p_{\min}} + \frac{75L(1-K\mu_t)\ln(N)}{\mu_t^2 t p_{\min}^2}.$$

The theorem now follows from the bounds of Eq. (8). $\qquad\qquad\square$

The other main deviation bound is a straightforward application of Freedman's inequality and a union bound. To state the lemma, we must introduce one more definition. Let

$$\mathcal{V}_t(\pi) := \max_{0\leq\tau\leq t-1} V(\tilde{Q}_\tau, \pi, \mu_\tau)$$

where $\tilde{Q}_\tau$ is the distribution calculated in Step 3 of Algorithm 1. Note that $\tilde{Q}_\tau^{\mu_\tau}$ is the distribution used to select the composite action in round $\tau + 1$.

**Lemma 7.** *Let $\delta \in (0,1)$. Then with probability at least $1 - \delta/4$, for all $t \geq t_0$ and $\pi \in \Pi$, we have*

$$|\eta_t(\pi, w^\star) - \mathcal{R}(\pi)| \leq \frac{\|w^\star\|_2^2}{\|w^\star\|_1}\mathcal{V}_t(\pi)p_{\min}\mu_t + \frac{\|w^\star\|_2^2}{\|w^\star\|_1}KL\mu_t. \qquad (11)$$

*Proof.* Consider a specific $t \geq t_0$ and $\pi \in \Pi$. Let

$$Z_i \coloneqq \langle w^\star, \hat{y}_i(\pi(x_i)) \rangle - \langle w^\star, y_i(\pi(x_i)) \rangle$$

and note that $\frac{1}{t} \sum_{i=1}^{t} Z_i = \eta_t(\pi, w^\star) - \mathcal{R}(\pi)$. Since $\hat{y}_i$ is an unbiased estimate of $y_i$, the $Z_i$s form a martingale. The range of each $Z_i$ is bounded as

$$|Z_i| \leq \|w^\star\|_1 \|\hat{y}_i - y_i\|_\infty \leq \frac{\|w^\star\|_1}{\mu_{i-1} p_{\min}} \leq \frac{\|w^\star\|_1}{\mu_t p_{\min}},$$

because the $\mu_i$s are non-increasing. The conditional variance can be bounded via the Cauchy-Schwarz inequality:

$$\mathbb{E}[Z_i^2 \mid H_{i-1}] \leq \|w^\star\|_2^2 \sum_{\ell=1}^{L} \mathbb{E}_{x \sim \mathcal{D}_x} \mathbb{E}_{y|x} \frac{y(\pi(x)_\ell)^2}{\tilde{Q}_{i-1}^{\mu_{i-1}}(\pi(x)_\ell \mid x)}$$

$$\leq \|w^\star\|_2^2 V(\tilde{Q}_{i-1}, \pi, \mu_{i-1}) \leq \|w^\star\|_2^2 \mathcal{V}_t(\pi).$$

By Freedman's inequality with $\lambda = \mu_t p_{\min} / \|w^\star\|_1$, we have, with probability at least $1 - \delta/(8t^2 N)$,

$$|\eta_t(\pi, w^\star) - \mathcal{R}(\pi)| \leq \frac{\mu_t p_{\min}}{\|w^\star\|_1} \cdot \|w^\star\|_2^2 \mathcal{V}_t(\pi) + \frac{d_t}{t} \cdot \frac{\|w^\star\|_1}{\mu_t p_{\min}}$$

$$\leq \frac{\|w^\star\|_2^2}{\|w^\star\|_1} \mathcal{V}_t(\pi) p_{\min} \mu_t + K \mu_t \|w^\star\|_1 \tag{12}$$

$$\leq \frac{\|w^\star\|_2^2}{\|w^\star\|_1} \mathcal{V}_t(\pi) p_{\min} \mu_t + \frac{\|w^\star\|_2^2}{\|w^\star\|_1} KL \mu_t. \tag{13}$$

Here, Eq. (12) follows because $d_t/(p_{\min} t) \leq K \mu_t^2$ by Eq. (8). Eq. (13) follows because $\|w^\star\|_1 \leq L \|w^\star\|_2^2 / \|w^\star\|_1$ by the fact that $\|w^\star\|_1 \leq \sqrt{L} \|w^\star\|_2$. The lemma follows by a union bound over all $t \geq t_0$ and $\pi \in \Pi$. $\qquad\square$

Equipped with these two deviation bounds we will proceed to prove the main theorem. Let $\mathcal{E}$ denote the event that both the variance and reward deviation bounds of Theorem 5 and Lemma 7 hold. Note that $\mathbb{P}(\mathcal{E}) \geq 1 - \delta/2$. Using the variance constraint, it is straightforward to prove the following lemma:

**Lemma 8.** *Assume event $\mathcal{E}$ holds, then for any round $t \geq 1$ and any policy $\pi \in \Pi$, let $t_\star$ be the round achieving the $\max$ in the definition of $\mathcal{V}_t(\pi)$. Then there are universal constants $\theta_1 \geq 2$ and $\theta_2$ such that:*

$$\mathcal{V}_t(\pi) \leq \begin{cases} \dfrac{2KL}{p_{\min}} & \text{if } t_\star < t_0, \\[2mm] \dfrac{\theta_1 KL}{p_{\min}} + \dfrac{\|w^\star\|_1}{\|w^\star\|_2^2} \dfrac{\widehat{\text{Reg}}_{t_\star}(\pi)}{\theta_2 p_{\min} \mu_{t_\star}} & \text{if } t_\star \geq t_0. \end{cases} \tag{14}$$

*Proof.* The first claim follows by the definition of $\mathcal{V}_t(\pi)$ and the fact that $\mu_\tau = 1/2K$ for $\tau < t_0$. For the second claim, we use the variance deviation bound and the optimization constraint. In particular, since $t_\star \geq t_0$, we can apply Theorem 5:

$$V(\tilde{Q}_{t_\star}, \pi, \mu_{t_\star}) \leq 6.4 \hat{V}_{t_\star}(\tilde{Q}_{t_\star}, \pi, \mu_{t_\star}) + 81.3 \frac{KL}{p_{\min}},$$

and we can use the optimization constraint which gives an upper bound on $\hat{V}_{t_\star}(\tilde{Q}_{t_\star}, \pi, \mu_{t_\star})$:

$$\hat{V}_{t_\star}(\tilde{Q}_{t_\star}, \pi, \mu_{t_\star}) \leq \hat{V}_{t_\star}(Q_{t_\star}, \pi, \mu_{t_\star}) \leq \frac{2KL}{p_{\min}} + \frac{\|w^\star\|_1}{\|w^\star\|_2^2} \frac{\widehat{\text{Reg}}_{t_\star}(\pi)}{\psi p_{\min} \mu_{t_\star}}$$

The bound follows by the choice $\theta_1 = 94.1$ and $\theta_2 = \psi/6.4$. $\qquad\square$

We next compare $\text{Reg}(\pi)$ and $\widehat{\text{Reg}}(\pi)$ using the variance bounds above.

**Lemma 9.** *Assume event $\mathcal{E}$ holds and define $c_0 := 4(1 + \theta_1)$. For all $t \geq t_0$ and all policies $\pi \in \Pi$:*

$$\operatorname{Reg}(\pi) \leq 2\widehat{\operatorname{Reg}}_t(\pi) + c_0 \frac{\|w^\star\|_2^2}{\|w^\star\|_1} KL\mu_t \quad and \quad \widehat{\operatorname{Reg}}_t(\pi) \leq 2\operatorname{Reg}(\pi) + c_0 \frac{\|w^\star\|_2^2}{\|w^\star\|_1} KL\mu_t. \quad (15)$$

*Proof.* The proof is by induction on $t$. As the base case, consider $t = t_0$ where we have $\mu_\tau = 1/(2K)$ for all $\tau < t_0$, so $\mathcal{V}_t(\pi) \leq 2KL/p_{\min}$ for all $\pi \in \Pi$ by Lemma 8. Using the reward deviation bound of Lemma 7, which holds under $\mathcal{E}$, we thus have

$$|\eta_t(\pi, w^\star) - \mathcal{R}(\pi)| \leq \frac{\|w^\star\|_2^2}{\|w^\star\|_1} \mathcal{V}_t(\pi) p_{\min}\mu_t + \frac{\|w^\star\|_2^2}{\|w^\star\|_1} KL\mu_t \leq 3\frac{\|w^\star\|_2^2}{\|w^\star\|_1} KL\mu_t$$

for all $\pi \in \Pi$. Now both directions of the bound follow from the triangle inequality and the optimality of $\pi_t$ for $\eta_t(\cdot)$ and $\pi_\star$ for $\mathcal{R}(\cdot)$, using the fact that $c_0 \geq 6$ from the definition of $\theta_1$.

For the inductive step, fix some round $t$ and assume that the claim holds for all $t_0 \leq t' < t$ and all $\pi \in \Pi$. By the optimality of $\pi_t$ for $\eta_t$ and Lemma 7, we have

$$\begin{aligned}
\operatorname{Reg}(\pi) - \widehat{\operatorname{Reg}}_t(\pi) &= (\mathcal{R}(\pi_\star) - \mathcal{R}(\pi)) - (\eta_t(\pi_t, w^\star) - \eta_t(\pi, w^\star)) \\
&\leq (\mathcal{R}(\pi_\star) - \mathcal{R}(\pi)) - (\eta_t(\pi_\star, w^\star) - \eta_t(\pi, w^\star)) \\
&\leq (\mathcal{V}_t(\pi_\star) + \mathcal{V}_t(\pi))\frac{\|w^\star\|_2^2}{\|w^\star\|_1} p_{\min}\mu_t + 2\frac{\|w^\star\|_2^2}{\|w^\star\|_1} KL\mu_t.
\end{aligned}$$

Now by Lemma 8, there exist rounds $i, j < t$ such that

$$\mathcal{V}_t(\pi) \leq \frac{\theta_1 KL}{p_{\min}} + \frac{\|w^\star\|_1}{\|w^\star\|_2^2} \frac{\widehat{\operatorname{Reg}}_i(\pi)}{\theta_2 p_{\min}\mu_i} \mathbf{1}(i \geq t_0)$$

$$\mathcal{V}_t(\pi_\star) \leq \frac{\theta_1 KL}{p_{\min}} + \frac{\|w^\star\|_1}{\|w^\star\|_2^2} \frac{\widehat{\operatorname{Reg}}_j(\pi_\star)}{\theta_2 p_{\min}\mu_j} \mathbf{1}(j \geq t_0)$$

For the term involving $\mathcal{V}_t(\pi)$, if $i < t_0$, we immediately have the bound

$$\mathcal{V}_t(\pi)\frac{\|w^\star\|_2^2}{\|w^\star\|_1} p_{\min}\mu_t \leq \theta_1 \frac{\|w^\star\|_2^2}{\|w^\star\|_1} KL\mu_t.$$

On the other hand, if $i \geq t_0$ then using the fact that $\mu_i \geq \mu_t$, and applying the inductive hypothesis to $\widehat{\operatorname{Reg}}_i(\pi)$ gives:

$$\mathcal{V}_t(\pi)\frac{\|w^\star\|_2^2}{\|w^\star\|_1} p_{\min}\mu_t \leq \theta_1 \frac{\|w^\star\|_2^2}{\|w^\star\|_1} KL\mu_t + \frac{\widehat{\operatorname{Reg}}_i(\pi)\mu_t}{\theta_2\mu_i} \leq \left(\theta_1 + \frac{c_0}{\theta_2}\right)\frac{\|w^\star\|_2^2}{\|w^\star\|_1} KL\mu_t + \frac{2\operatorname{Reg}(\pi)}{\theta_2}.$$

Similarly for the $\mathcal{V}_t(\pi_\star)$ term, we have the bound

$$\mathcal{V}_t(\pi_\star)\frac{\|w^\star\|_2^2}{\|w^\star\|_1} p_{\min}\mu_t \leq \left(\theta_1 + \frac{c_0}{\theta_2}\right)\frac{\|w^\star\|_2^2}{\|w^\star\|_1} KL\mu_t + \frac{2\operatorname{Reg}(\pi_\star)}{\theta_2} = \left(\theta_1 + \frac{c_0}{\theta_2}\right)\frac{\|w^\star\|_2^2}{\|w^\star\|_1} KL\mu_t,$$

since $\pi_\star$ has no regret. Combining these bounds gives:

$$\operatorname{Reg}(\pi) - \widehat{\operatorname{Reg}}_t(\pi) \leq 2\left(\theta_1 + \frac{c_0}{\theta_2}\right)\frac{\|w^\star\|_2^2}{\|w^\star\|_1} KL\mu_t + \frac{2\operatorname{Reg}(\pi)}{\theta_2} + 2\frac{\|w^\star\|_2^2}{\|w^\star\|_1} KL\mu_t,$$

which gives

$$\operatorname{Reg}(\pi) \leq \frac{1}{1 - 2/\theta_2}\left(\widehat{\operatorname{Reg}}_t(\pi) + 2\left(1 + \theta_1 + \frac{c_0}{\theta_2}\right)\frac{\|w^\star\|_2^2}{\|w^\star\|_1} KL\mu_{t-1}\right).$$

Recall that $\theta_1 = 94.1, \theta_2 = \psi/6.4, \psi = 100$, and $c_0 = 4(1 + \theta_1)$. This means that $\theta_2 > 15.6$, so $2/\theta_2 \leq 1/2$, and hence the pre-multiplier on the $\widehat{\operatorname{Reg}}_t(\pi)$ term is at most 2. To finish proving the bound on $\operatorname{Reg}(\pi)$, it remains to show that $c_0 \geq 2(1 + \theta_1 + c_0/\theta_2)/(1 - 2/\theta_2)$, or equivalently, that

$$c_0 (1 - 4/\theta_2) \geq 2 (1 + \theta_1).$$

This holds, because $c_0(1 - 4/\theta_2) = 4(1 - 4/\theta_2)(1 + \theta_1)$ and $4/\theta_2 \leq 1/2$.

The other direction proceeds similarly. Under event $\mathcal{E}$ we have:

$$\widehat{\mathrm{Reg}}_t(\pi) - \mathrm{Reg}(\pi) = \eta_t(\pi_t, w^\star) - \eta_t(\pi, w^\star) - \mathcal{R}(\pi_\star) + \mathcal{R}(\pi)$$
$$\leq \eta_t(\pi_t, w^\star) - \eta_t(\pi, w^\star) - \mathcal{R}(\pi_t) + \mathcal{R}(\pi)$$
$$\leq (\mathcal{V}_t(\pi) + \mathcal{V}_t(\pi_t)) \frac{\|w^\star\|_2^2}{\|w^\star\|_1} p_{\min} \mu_t + 2 \frac{\|w^\star\|_2^2}{\|w^\star\|_1} K L \mu_t.$$

As before, we have the bound:

$$\mathcal{V}_t(\pi) \frac{\|w^\star\|_2^2}{\|w^\star\|_1} p_{\min} \mu_t \leq \left( \theta_1 + \frac{c_0}{\theta_2} \right) \frac{\|w^\star\|_2^2}{\|w^\star\|_1} K L \mu_t + \frac{2\mathrm{Reg}(\pi)}{\theta_2},$$

but for the $\mathcal{V}_t(\pi_t)$ term we must use the inductive hypothesis twice. We know there exists a round $j < t$ for which

$$\mathcal{V}_t(\pi_t) \leq \theta_1 \frac{K L}{p_{\min}} + \frac{\|w^\star\|_1}{\|w^\star\|_2^2} \frac{\widehat{\mathrm{Reg}}_j(\pi)}{\theta_2 p_{\min} \mu_j} \mathbf{1}(j \geq t_0).$$

Applying the inductive hypothesis twice gives:

$$\frac{\|w^\star\|_1}{\|w^\star\|_2^2} \frac{\widehat{\mathrm{Reg}}_j(\pi_t)}{\theta_2 p_{\min} \mu_j} \leq \frac{\|w^\star\|_1}{\|w^\star\|_2^2} \frac{\left( 2\mathrm{Reg}(\pi_t) + c_0 \frac{\|w^\star\|_2^2}{\|w^\star\|_1} K L \mu_j \right)}{\theta_2 p_{\min} \mu_j}$$
$$\leq \frac{\|w^\star\|_1}{\|w^\star\|_2^2} \frac{2 \left( 2\widehat{\mathrm{Reg}}_t(\pi_t) + c_0 \frac{\|w^\star\|_2^2}{\|w^\star\|_1} K L \mu_t \right) + c_0 \frac{\|w^\star\|_2^2}{\|w^\star\|_1} K L \mu_j}{\theta_2 p_{\min} \mu_j}$$
$$\leq \frac{3c_0}{\theta_2} \frac{K L}{p_{\min}}.$$

Here we use the inductive hypothesis twice, once at round $j$ and once at round $t$, and then use the fact that $\pi_t$ has no regret at round $t$, i.e., $\widehat{\mathrm{Reg}}_t(\pi_t) = 0$. We also use the fact that the $\mu_t$s are non-increasing, so $\mu_t/\mu_j \leq 1$. This gives the bound:

$$\mathcal{V}_t(\pi_t) \frac{\|w^\star\|_2^2}{\|w^\star\|_1} p_{\min} \mu_t \leq \left( \theta_1 + \frac{3c_0}{\theta_2} \right) \frac{\|w^\star\|_2^2}{\|w^\star\|_1} K L \mu_t.$$

Combining the bounds for $\mathcal{V}_t(\pi)$ and $\mathcal{V}_t(\pi_t)$ gives:

$$\widehat{\mathrm{Reg}}_t(\pi) \leq \left( 1 + \frac{2}{\theta_2} \right) \mathrm{Reg}(\pi) + \left( 2\theta_1 + \frac{4c_0}{\theta_2} + 2 \right) \frac{\|w^\star\|_2^2}{\|w^\star\|_1} K L \mu_t.$$

Since $\theta_2 \geq 2$, the pre-multiplier on the first term is at most 2. It remains to show that $c_0 \geq 2(1+\theta_1) + 4c_0/\theta_2$. This is again equivalent to $c_0(1 - 4/\theta_2) \geq 2(1+\theta_1)$, which holds as before. $\square$

The last key ingredient of the proof is the following lemma, which shows that the low-regret constraint in Eq. (4), based on the regret estimates, actually ensures low regret.

**Lemma 10.** *Assume event $\mathcal{E}$ holds. Then for every round $t \geq 1$:*

$$\sum_{\pi \in \Pi} \tilde{Q}_{t-1}(\pi) \mathrm{Reg}(\pi) \leq (4\psi + c_0) \frac{\|w^\star\|_2^2}{\|w^\star\|_1} K L \mu_{t-1} \qquad (16)$$

*Proof.* If $t \leq t_0$ then $\mu_{t-1} = 1/(2K)$ in which case (since $\mathrm{Reg}(\pi) \leq \|w^\star\|_1$):

$$\sum_{\pi \in \Pi} \tilde{Q}_{t-1}(\pi) \mathrm{Reg}(\pi) \leq \|w^\star\|_1 \leq \frac{\|w^\star\|_2^2}{\|w^\star\|_1} L = 2 \frac{\|w^\star\|_2^2}{\|w^\star\|_1} K L \mu_{t-1} \leq (4\psi + c_0) \frac{\|w^\star\|_2^2}{\|w^\star\|_1} K L \mu_{t-1}.$$

For $t > t_0$, we have:

$$\sum_{\pi \in \Pi} \tilde{Q}_{t-1}(\pi) \mathrm{Reg}(\pi) \leq \sum_{\pi \in \Pi} \tilde{Q}_{t-1}(\pi) \left( 2\widehat{\mathrm{Reg}}_{t-1}(\pi) + c_0 \frac{\|w^\star\|_2^2}{\|w^\star\|_1} K L \mu_{t-1} \right)$$
$$\leq \left( 2 \sum_{\pi \in \Pi} Q_{t-1}(\pi) \widehat{\mathrm{Reg}}_{t-1}(\pi) \right) + c_0 \frac{\|w^\star\|_2^2}{\|w^\star\|_1} K L \mu_{t-1}$$
$$\leq (4\psi + c_0) \frac{\|w^\star\|_2^2}{\|w^\star\|_1} K L \mu_{t-1}.$$

The first inequality follows by Lemma 9 and the second follows from the fact that $\tilde{Q}_{t-1}$ places its remaining mass (compared with $Q_{t-1}$) on $\pi_{t-1}$ which suffers no empirical regret at round $t-1$. The last inequality is due to the low-regret constraint in the optimization. $\qquad\square$

To control the regret, we must first add up the $\mu_t$s, which relate to the exploration probability:

**Lemma 11.** *For any $T \geq 1$:*

$$\sum_{t=1}^{T} \mu_{t-1} \leq 2\sqrt{\frac{T d_T}{K p_{\min}}}.$$

*Proof.* We will use the identity

$$\frac{1}{K} \leq \sqrt{\frac{d_T}{K p_{\min}}}, \tag{17}$$

which holds, because $d_T \geq 1$ and $K \geq p_{\min}$. We prove the lemma separately for $T = 1$ and $T \geq 2$. Since $t_0 \geq 4$, we have $\mu_0 = 1/2K$. Thus, for $T = 1$, by Eq. (17):

$$\sum_{t=1}^{T} \mu_{t-1} = \frac{1}{2K} \leq \frac{1}{2}\sqrt{\frac{d_T}{K p_{\min}}} \leq 2\sqrt{\frac{T d_T}{K p_{\min}}}.$$

For $T \geq 2$, we use the fact that $\mu_0 = \mu_1 = 1/2K$, and $\mu_t \leq \sqrt{d_T/(K t p_{\min})}$ for $t \leq T$:

$$
\begin{aligned}
\sum_{t=1}^{T} \mu_{t-1} &\leq \frac{1}{K} + \sqrt{\frac{d_T}{K p_{\min}}} \sum_{t=3}^{T} \frac{1}{\sqrt{t-1}} \\
&\leq \sqrt{\frac{d_T}{K p_{\min}}} + \sqrt{\frac{d_T}{K p_{\min}}} \left(2\sqrt{T-1} - 2\right) \\
&\leq 2\sqrt{\frac{T d_T}{K p_{\min}}}.
\end{aligned}
\tag{18}
$$

In Eq. (18), we bounded the first term using Eq. (17) and the second term using the telescoping identity $1/\sqrt{t-1} \leq 2\sqrt{t-1} - 2\sqrt{t-2}$, which holds for $t \geq 2$. $\qquad\square$

We are finally ready to prove the theorem by adding up the total regret for the algorithm.

**Lemma 12.** *For any $T \in \mathbb{N}$, with probability at least $1 - \delta$, the regret after $T$ rounds is at most:*

$$\frac{\|w^\star\|_2^2}{\|w^\star\|_1} L \left[ 2\sqrt{2T \ln(2/\delta)} + 2(4\psi + c_0 + 1)\sqrt{\frac{K T d_T}{p_{\min}}} \right].$$

*Proof.* For each round $t \geq 1$, let $Z_t := r_t(\pi_\star(x_t)) - r_t(A_t) - \sum_{\pi \in \Pi} \tilde{Q}_{t-1}^{\mu_{t-1}}(\pi) \operatorname{Reg}(\pi)$. Since at round $t$, we play action $A_t$ with probability $\tilde{Q}_{t-1}^{\mu_{t-1}}(A_t)$, we have $\mathbb{E}Z_t = 0$. Moreover, since the noise term $\xi$ is shared between $r_t(\pi_\star(x_t))$ and $r_t(A_t)$, we have $|Z_i| \leq 2\|w^\star\|_1$ and it follows by Azuma's inequality (Lemma 24) that with probability at least $1 - \delta/2$:

$$\sum_{t=1}^{T} |Z_t| \leq 2\|w^\star\|_1 \sqrt{2T \ln(2/\delta)}.$$

To control the mean, we use event $\mathcal{E}$, which, by Theorem 5 and Lemma 7, holds with probability at least $1 - \delta/2$. By another union bound, with probability at least $1 - \delta$, the regret of the algorithm is

---

**Algorithm 4** Coordinate Ascent Algorithm for Semi-Bandit Optimization Problem (OP)

---

**Require:** History $H$ and smoothing parameter $\mu$.

1: Initialize weights $Q \leftarrow 0 \in \Delta_{\leq}(\Pi)$.
2: **while** true **do**
3:  For all $\pi$, define:

$$V_\pi(Q) := \hat{\mathbb{E}}_{x \sim H}\left[\sum_{\ell=1}^{L} \frac{1}{Q^\mu(\pi(x)_\ell|x)}\right], \qquad S_\pi(Q) := \hat{\mathbb{E}}_{x \sim H}\left[\sum_{\ell=1}^{L} \frac{1}{Q^\mu(\pi(x)_\ell|x)^2}\right],$$

$$D_\pi(Q) := V_\pi(Q) - \frac{2KL}{p_{\min}} - b_\pi$$

4:  If $\sum_\pi Q(\pi)(\frac{2KL}{p_{\min}} + b_\pi) > \frac{2KL}{p_{\min}}$, replace $Q$ by $cQ$ where $c := \frac{2KL/p_{\min}}{\sum_\pi Q(\pi)(2KL/p_{\min}+b_\pi)} < 1$.
5:  Else if $\exists \pi$ s.t. $D_\pi(Q) > 0$, update $Q(\pi) \leftarrow Q(\pi) + \alpha_\pi(Q)$ where $\alpha_\pi(Q) := \frac{V_\pi(Q)+D_\pi(Q)}{2(1-K\mu)S_\pi(Q)}$.
6:  Otherwise halt and output $Q$.
7: **end while**

---

bounded by:

$$\text{Regret} \leq 2\|w^\star\|_1 \sqrt{2T \ln(2/\delta)} + \sum_{t=1}^{T} \sum_{\pi \in \Pi} \tilde{Q}_{t-1}^{\mu_{t-1}}(\pi)\text{Reg}(\pi)$$

$$\leq 2\|w^\star\|_1 \sqrt{2T \ln(2/\delta)} + \sum_{t=1}^{T} \sum_{\pi \in \Pi} \left[(1 - K\mu_{t-1})\tilde{Q}_{t-1}(\pi)\text{Reg}(\pi) + \|w^\star\|_1 K\mu_{t-1}\right]$$

$$\leq 2\|w^\star\|_1 \sqrt{2T \ln(2/\delta)} + \sum_{t=1}^{T} (4\psi + c_0 + 1)\frac{\|w^\star\|_2^2}{\|w^\star\|_1} LK\mu_{t-1}$$

$$\leq \frac{\|w^\star\|_2^2}{\|w^\star\|_1} L \left[2\sqrt{2T \ln(2/\delta)} + 2(4\psi + c_0 + 1)\sqrt{\frac{KTd_T}{p_{\min}}}\right]$$

Here the first inequality is from the application of Azuma's inequality above. The second one uses the definition of $\tilde{Q}_{t-1}^{\mu_{t-1}}$ to split into rounds where we play as $\tilde{Q}_{t-1}$ and rounds where we explore. The exploration rounds occur with probability $K\mu_{t-1}$, and on those rounds we suffer regret at most $\|w^\star\|_1$. For the other rounds, we use Lemma 10 and then Lemma 11. We collect terms using the inequality $\|w^\star\|_1 \leq L\|w^\star\|_2^2/\|w^\star\|_1$. $\qquad \square$

## E  Proof of Oracle Complexity Bound in Theorem 1

In this section we prove the oracle complexity bound in Theorem 1. First we describe how the optimization problem OP can be solved via a coordinate ascent procedure. Similar to the previous appendix, we use the shorthand $Q(a \mid x)$ to mean $Q(a \in A \mid x)$ for any projected subdistribution $Q(A \mid x)$. If $Q$ is a distribution, we have $\sum_{a \in \mathcal{A}} Q(a \mid x) = L$. For a subdistribution, this number can be smaller.

This problem is similar to the one used by Agarwal et al. [1] for contextual bandits rather than semibandits, and following their approach, we provide a coordinate ascent procedure in the policy space (see Algorithm 4). There are two types of updates in the algorithm. If the weights $Q$ are too large or the regret constraint in Equation 4 is violated, the algorithm multiplicatively shrinks all of the weights. Otherwise, if there is a policy that is found to violate the variance constraint in Equation 5, the algorithm adds weight to that policy, so that the constraint is no longer violated.

First, if the algorithm halts, then both of the conditions must be satisfied. The regret condition must be satisfied since we know that $\sum_\pi Q(\pi)(2KL/p_{\min} + b_\pi) \leq 2KL/p_{\min}$ which in particular implies that $\sum_\pi Q(\pi)b_\pi \leq 2KL/p_{\min}$ as required. Note that this also ensures that $\sum_\pi Q(\pi) \leq 1$

so $Q \in \Delta_{\leq}(\Pi)$. Finally, if we halted, then for each $\pi$, we must have $D_\pi(Q) \leq 0$ which implies $V_\pi(Q) \leq \frac{2KL}{p_{\min}} + b_\pi$ so the variance constraint is also satisfied.

The algorithm can be implemented by first accessing the oracle on the importance weighted history $\{(x_\tau, \hat{y}_\tau, w^\star)\}_{\tau=1}^t$ at the end of round $t$ to obtain $\pi_t$, which we also use to compute $b_\pi$. The low regret check in Step 4 of Algorithm 4 can be done efficiently, since each policy in the support of the current distribution $Q$ was added at a previous iteration of Algorithm 4, and we can store the regret of the policy at that time for no extra computational burden. This allows us to always maintain the expected regret of the current distribution $Q$ for no added cost. Finding a policy violating the variance check can be done by one call to the AMO. At round $t$, we create a dataset of the form $(x_i, z_i, v_i)$ of size $2t$. The first $t$ terms come from the variance $V_\pi(Q)$ and the second $t$ terms come from the rescaled empirical regret $b_\pi$. For $\tau \leq t$, we define $x_\tau$ to be the $\tau^{\text{th}}$ context,

$$z_\tau(a) := \frac{1}{t Q^\mu(a|x_\tau)}, \quad \text{and} \quad v_\tau := \mathbf{1}.$$

With this definition, it is easily seen that $V_\pi(Q) = \sum_{\tau=1}^t v_\tau^T z_\tau(\pi(x_\tau))$. For $\tau > t$, we define $x_\tau$ to be the context from round $\tau - t$ and

$$z_\tau(a) := \frac{-\|w^\star\|_1}{\|w^\star\|_2^2 t \psi \mu p_{\min}} \hat{y}_\tau(a), \quad \text{and} \quad v_\tau := w^\star.$$

It can now be verified that $\sum_{\tau=t+1}^{2t} v_\tau^T z_\tau$ recovers the $b_\pi$ term up to additive constants independent of the policy $\pi$ (essentially up to the $\eta_t(\pi_t)$ term). Combining everything, it can be checked that:

$$D_\pi(Q) = \sum_{\tau=1}^{2t} \langle z_\tau(\pi(x_\tau)), v_\tau \rangle - \frac{2KL}{p_{\min}} - \frac{\|w^\star\|_1}{\|w^\star\|_2^2} \frac{\eta_t(\pi_t)}{\psi \mu p_{\min}}$$

The two terms at the end are independent of $\pi$ so by calling the argmax oracle with this $2t$ sized dataset, we can find the policy $\pi$ with the largest value of $D_\pi$. If the largest value is non-positive, then no constraint violation exists. If it is strictly positive, then we have found a constraint violator that we use to update the probability distribution.

As for the iteration complexity, we prove the following theorem.

**Theorem 13.** *For any history $H$ and parameter $\mu$, Algorithm 4 halts and outputs a set of weights $Q \in \Delta_{\leq}(\Pi)$ that is feasible for* OP. *Moreover, Algorithm 4 halts in no more than $\frac{8 \ln(1/(K\mu))}{\mu p_{\min}}$ iterations and each iteration can be implemented efficiently, with at most one call to* AMO.

Equipped with this theorem, it is easy to see that the total number of calls to the AMO over the course of the execution of Algorithm 1 can be bounded as $\tilde{O}\left(T^{3/2}\sqrt{\frac{K}{p_{\min}\log(N/\delta)}}\right)$ by the setting of $\mu_t$. Moreover, due to the nature of the coordinate ascent algorithm, the weight vector $Q$ remains sparse, so we can manipulate it efficiently and avoid running time that is linear in $N$. As mentioned, this contrasts with the exponential-weights style algorithm of Kale et al. [12] which maintains a dense weight vector over $\Delta_{\leq}(\Pi)$.

We mention in passing that Agarwal et al. [1] also develop two improvements that lead to a more efficient algorithm. They partition the learning process into epochs and only solve OP once every epoch, rather than in every round as we do here (Lemma 2 in Agarwal et al. [1]). They also show how to use the weight vector from the previous round to warm-start the next coordinate ascent execution (Lemma 3 in Agarwal et al. [1]). Both of these optimizations can also be implemented here, and we expect they will reduce the total number of oracle calls over $T$ rounds to scale with $\sqrt{T}$ rather than $T^{3/2}$ as in our result. We omit these details to simplify the presentation.

### E.1 Proof of Theorem 13

Throughout the proof we write $U(A \mid x)$ instead of $U_x(A)$ to parallel the notation $Q(A \mid x)$. Also, similarly to $Q(a \mid x)$, we write $U(a \mid x)$ to mean $U_x(a \in A)$.

We use the following potential function for the analysis, which is adapted from Agarwal et al. [1],

$$\Phi(Q) := \frac{\hat{\mathbb{E}}_{x \sim H}\left[\mathrm{RE}\big(U(\cdot \mid x) \,\|\, Q^\mu(\cdot \mid x)\big)\right]}{1 - K\mu} + \frac{\sum_\pi Q(\pi) b_\pi}{2K/p_{\min}}$$

with

$$\mathrm{RE}(p\|q) \coloneqq \sum_{a \in \mathcal{A}} p_a \ln(p_a/q_a) + q_a - p_a$$

being the unnormalized relative entropy. Its arguments $p$ and $q$ can be any non-negative vectors in $\mathbb{R}^K$. For intuition, note that the partial derivative of the potential function with respect to a coordinate $Q(\pi)$ relates to the variance $V_\pi(Q)$ as follows:

$$\frac{\partial \Phi(Q)}{\partial Q(\pi)} = \frac{\hat{\mathbb{E}}_{x \sim H}\left[\sum_{a \in \pi(x)}\left(-\frac{U(a|x)}{Q^\mu(a|x)}(1 - K\mu) + (1 - K\mu)\right)\right]}{1 - K\mu} + \frac{b_\pi}{2K/p_{\min}}$$

$$= -\hat{\mathbb{E}}_{x \sim H}\left[\sum_{a \in \pi(x)} \frac{U(a \mid x)}{Q^\mu(a \mid x)}\right] + L + \frac{p_{\min} b_\pi}{2K}$$

$$\leq -\frac{p_{\min}}{K} V_\pi(Q) + L + \frac{p_{\min} b_\pi}{2K}$$

$$= \frac{p_{\min}}{2K}\left(-2V_\pi(Q) + \frac{2KL}{p_{\min}} + b_\pi\right)$$

$$= \frac{p_{\min}}{2K}\left(-D_\pi(Q) - V_\pi(Q)\right).$$

This means that if $D_\pi(Q) > 0$, then the partial derivative is very negative, and by increasing the weight $Q(\pi)$, we can decrease the potential function $\Phi$.

We establish the following five facts:

1. $\Phi(0) \leq L \ln(1/(K\mu))/(1 - K\mu)$.
2. $\Phi(Q)$ is convex in $Q$.
3. $\Phi(Q) \geq 0$ for all $Q$.
4. The shrinking update, when the regret constraint is violated, does not increase the potential. More formally, for any $c < 1$, we have $\Phi(cQ) \leq \Phi(Q)$ whenever $\sum_\pi Q(\pi)(2KL/p_{\min} + b_\pi) > 2KL/p_{\min}$.
5. The additive update, when $D_\pi > 0$ for some $\pi$, lowers the potential by at least $\frac{L\mu p_{\min}}{4(1 - K\mu)}$.

With these five facts, establishing the result is straightforward. In every iteration, we either terminate, perform the shrinking update, or the additive update. However, we will never perform the shrinking update in two consecutive iterations, since our choice of $c$, ensures the condition is not satisfied in the next iteration. Thus, we perform the additive update at least once every two iterations. If we perform $I$ iterations, by the fifth fact, we are guaranteed to decrease the potential $\Phi$ by,

$$\frac{I}{2} \frac{L\mu p_{\min}}{4(1 - K\mu)} = \frac{IL\mu p_{\min}}{8(1 - K\mu)}$$

However, the total change in potential is bounded by $L \ln(1/(K\mu))/(1 - K\mu)$ by the first and second facts. Thus, we must have

$$\frac{IL\mu p_{\min}}{8(1 - K\mu)} \leq \frac{L \ln(1/(K\mu))}{(1 - K\mu)},$$

which is precisely the claim.

We now turn to proving the five facts. The first three are fairly straightforward and the last two follow from analogous claims as in Agarwal et al. [1]. To prove the first fact, note that the exploration distribution in $Q^\mu$ is exactly $U_x$, so

$$\Phi(0) = \hat{\mathbb{E}}_{x \sim H}\left[\sum_{a \in \mathcal{A}} \frac{U(a \mid x) \ln\left(\frac{U(a|x)}{K\mu U(a|x)}\right) - (1 - K\mu)U(a \mid x)}{1 - K\mu}\right] \leq \frac{L \ln(1/(K\mu))}{1 - K\mu},$$

because $\sum_{a\in\mathcal{A}} U(a\,|\,x) = L$ since $U(A\,|\,x)$ is a distribution. Convexity of this function follows from the fact that the unnormalized relative entropy is convex in the second argument, and the fact that the weight vector $q \in \mathbb{R}^K$ with components $q_a = Q^\mu(a\,|\,x)$ is a linear transformation of $Q \in \mathbb{R}^N$. The third fact follows by the non-negativity of both the empirical regret $b_\pi$ and the unnormalized relative entropy $\mathrm{RE}(\cdot\|\cdot)$. For the fourth fact, we prove the following lemma.

**Lemma 14.** *Let $Q$ be a weight vector for which $\sum_\pi Q(\pi)(2KL/p_{\min} + b_\pi) > 2KL/p_{\min}$ and define $c := \frac{2KL/p_{\min}}{\sum_\pi Q(\pi)(2KL/p_{\min}+b_\pi)} < 1$. Then $\Phi(cQ) \leq \Phi(Q)$.*

*Proof.* Let $g(c) := \Phi(cQ)$ and $Q_c^\mu(a|x) := (1 - K\mu)cQ(a|x) + K\mu U(a|x)$. By the chain rule, using the calculation of the derivative above, we have:

$$g'(c) = \sum_\pi Q(\pi) \frac{\partial \Phi(cQ)}{\partial Q(\pi)}$$

$$= \frac{p_{\min}}{2K} \sum_\pi Q(\pi)\left(\frac{2KL}{p_{\min}} + b_\pi\right) - \sum_\pi Q(\pi)\hat{\mathbb{E}}_x\left[\sum_{a\in\pi(x)} \frac{U(a|x)}{Q_c^\mu(a|x)}\right]. \tag{19}$$

Analyze the last term:

$$\sum_\pi Q(\pi)\hat{\mathbb{E}}_x\left[\sum_{a\in\pi(x)} \frac{U(a|x)}{Q_c^\mu(a|x)}\right] = \hat{\mathbb{E}}_x\left[\sum_{a\in\mathcal{A}}\sum_{\pi\in\Pi} \frac{U(a|x)Q(\pi)\mathbf{1}(a \in \pi(x))}{Q_c^\mu(a|x)}\right]$$

$$= \hat{\mathbb{E}}_x\left[\sum_{a\in\mathcal{A}} \frac{U(a|x)Q(a|x)}{Q_c^\mu(a|x)}\right] = \frac{1}{c}\hat{\mathbb{E}}_x\left[\sum_{a\in\mathcal{A}} \frac{U(a|x)cQ(a|x)}{Q_c^\mu(a|x)}\right]. \tag{20}$$

We now focus on one context $x$ and define $q_a := cQ(a|x)$ and $u_a := U(a|x)/L$. Note that $\sum_a U(a|x) = L$ so the vector $u$ describes a probability distribution over $a \in \mathcal{A}$. The inner sum in Eq. (20) can be upper bounded by:

$$\sum_{a\in\mathcal{A}} \frac{U(a|x)cQ(a|x)}{Q_c^\mu(a|x)} = \sum_{a\in\mathcal{A}} \frac{Lu_a q_a}{(1-K\mu)q_a + KL\mu u_a} = \sum_{a\in\mathcal{A}} \frac{Lu_a(q_a/u_a)}{(1-K\mu)(q_a/u_a) + KL\mu}$$

$$= L\mathbb{E}_{a\sim u}\left[\frac{q_a/u_a}{(1-K\mu)(q_a/u_a) + KL\mu}\right]$$

$$\leq \frac{L\mathbb{E}_{a\sim u}[q_a/u_a]}{(1-K\mu)\mathbb{E}_{a\sim u}[q_a/u_a] + KL\mu}$$

$$= \frac{L(\sum_{a\in\mathcal{A}} q_a)}{(1-K\mu)(\sum_{a\in\mathcal{A}} q_a) + KL\mu}$$

$$\leq \frac{L^2}{(1-K\mu)L + KL\mu} = L. \tag{21}$$

In the third line we use Jensen's inequality, noting that $x/(ax + b)$ is concave in $x$ for $a \geq 0$. In Eq. (21), we use that $\sum_{a\in\mathcal{A}} q_a \leq L$ and that $x/(ax + b)$ is non-decreasing, so plugging in $L$ for $\sum_a q_a$ gives an upper bound.

Combining Eqs. (19), (20), and (21), and plugging in our choice of $c = \frac{2KL/p_{\min}}{\sum_\pi Q(\pi)(2KL/p_{\min}+b_\pi)}$, we obtain the following lower bound on $g'(c)$:

$$g'(c) \geq \frac{p_{\min}}{2K}\sum_\pi Q(\pi)\left(\frac{2KL}{p_{\min}} + b_\pi\right) - \frac{L}{c}$$

$$= \frac{p_{\min}}{2K}\left(\sum_\pi Q(\pi)\left(\frac{2KL}{p_{\min}} + b_\pi\right) - \frac{2KL}{cp_{\min}}\right) = 0.$$

Since $g$ is convex, this means that $g(c')$ is nondecreasing for all values $c'$ exceeding $c$. Since $c < 1$, we have:

$$\Phi(Q) = g(1) \geq g(c) = \Phi(cQ). \qquad \square$$

And for the fifth fact, we have:

**Lemma 15.** *Let $Q$ be a subdistribution and suppose, for some policy $\pi$, that $D_\pi(Q) > 0$. Let $Q'$ be the new set of weights which is identical except that $Q'(\pi) := Q(\pi) + \alpha$ with $\alpha := \alpha_\pi(Q) > 0$. Then*

$$\Phi(Q) - \Phi(Q') \geq \frac{L\mu p_{\min}}{4(1 - K\mu)}.$$

*Proof.* Assume $D_\pi(Q) > 0$. Note that the updated subdistribution equals $Q'(\cdot) = Q(\cdot) + \alpha \mathbf{1}(\cdot = \pi)$, so its smoothed projection, $Q'^\mu(a \mid x) = Q^\mu(a \mid x) + (1 - K\mu)\alpha \mathbf{1}(a \in \pi(x))$, differs only in a small number of coordinates from $Q^\mu(a \mid x)$. Using the shorthand $q_a^\mu := Q^\mu(a \mid x)$, $q_a'^\mu := Q'^\mu(a \mid x)$ and $u_a := U(a \mid x)$, we have:

$$2K\big(\Phi(Q) - \Phi(Q')\big) = 2K\left(\frac{\hat{\mathbb{E}}_x\left[\sum_a \left(u_a \ln(u_a/q_a^\mu) - u_a \ln(u_a/q_a'^\mu) + q_a^\mu - q_a'^\mu\right)\right]}{(1 - K\mu)} - \frac{\alpha b_\pi}{2K/p_{\min}}\right)$$

$$= \frac{2K}{1 - K\mu}\hat{\mathbb{E}}_x\left[\sum_{a \in \pi(x)} u_a \ln\left(\frac{q_a'^\mu}{q_a^\mu}\right)\right] - 2K\alpha L - \alpha b_\pi p_{\min}$$

$$\geq \frac{2p_{\min}}{1 - K\mu}\hat{\mathbb{E}}_x\left[\sum_{a \in \pi(x)} \ln\left(1 + \frac{\alpha(1 - K\mu)}{Q^\mu(a \mid x)}\right)\right] - p_{\min}\alpha\left(\frac{2KL}{p_{\min}} + b_\pi\right).$$

The term inside the expectation can be bounded using the fact that $\ln(1 + x) \geq x - x^2/2$ for $x \geq 0$:

$$\hat{\mathbb{E}}_x\left[\sum_{a \in \pi(x)} \ln\left(1 + \frac{\alpha(1 - K\mu)}{Q^\mu(a \mid x)}\right)\right] \geq \hat{\mathbb{E}}_x\left[\sum_{a \in \pi(x)}\left(\frac{\alpha(1 - K\mu)}{Q^\mu(a \mid x)} - \frac{\alpha^2(1 - K\mu)^2}{2Q^\mu(a \mid x)^2}\right)\right]$$

$$= \alpha(1 - K\mu)V_\pi(Q) - \frac{\alpha^2(1 - K\mu)^2}{2}S_\pi(Q).$$

Plugging this in the previous derivation gives a lower bound:

$$2K\big(\Phi(Q) - \Phi(Q')\big) \geq 2p_{\min}\alpha V_\pi(Q) - (1 - K\mu)p_{\min}\alpha^2 S_\pi(Q) - p_{\min}\alpha\left(\frac{2KL}{p_{\min}} + b_\pi\right)$$

$$\geq p_{\min}\alpha\big(V_\pi(Q) + D_\pi(Q)\big) - (1 - K\mu)p_{\min}\alpha^2 S_\pi(Q),$$

using the definition $D_\pi(Q) = V_\pi(Q) - \frac{2KL}{p_{\min}} - b_\pi$. Since $\alpha = \frac{V_\pi(Q) + D_\pi(Q)}{2(1 - K\mu)S_\pi(Q)}$, we obtain:

$$2K\big(\Phi(Q) - \Phi(Q')\big) \geq \frac{p_{\min}\big(V_\pi(Q) + D_\pi(Q)\big)^2}{4(1 - K\mu)S_\pi(Q)}$$

Note that $S_\pi(Q) \geq \frac{1}{\mu p_{\min}}V_\pi(Q)$ (by bounding the square terms in the definition of $S_\pi(Q)$ by a linear term times the lower bound, which is $\mu p_{\min}$) and that $V_\pi(Q) > \frac{2KL}{p_{\min}}$ since $D_\pi(Q) > 0$. Therefore:

$$2K\big(\Phi(Q) - \Phi(Q')\big) \geq \frac{\mu p_{\min}^2\big(V_\pi(Q) + D_\pi(Q)\big)^2}{4(1 - K\mu)V_\pi(Q)} \geq \frac{\mu p_{\min}^2 V_\pi(Q)}{4(1 - K\mu)} \geq \frac{KL\mu p_{\min}}{2(1 - K\mu)}.$$

Dividing both sides of this inequality by $2K$ proves the lemma. $\qquad\square$

# F   Proof of Theorem 2

The proof of Theorem 2 requires many delicate steps, so we first sketch the overall proof architecture. The first step is to derive a parameter estimation bound for learning in linear models. This is a somewhat standard argument from linear regression analysis, and the important component is that the bound involves the 2nd-moment matrix $\Sigma$ of the feature vectors used in the problem. Combining this with importance weighting on the reward features $y$ as in VCEE, we prove that the policy used in the exploitation phase has low expected regret, provided that $\Sigma$ has large eigenvalues.

The next step involves a precise characterization of the mean and deviation of the 2nd-moment matrix $\Sigma$, which relies on the exploration phase employing a uniform exploration strategy. This step involves a careful application of the matrix Bernstein inequality (Lemma 26). We then bound the expected regret accumulated during the exploration phase; we show, somewhat surprisingly, that the expected regret can be related to the mean of 2nd-moment matrix $\Sigma$ of the reward features. Finally, since per-round exploitation regret improves with a larger setting $\lambda_\star$, while the cumulative exploration regret improves with a smaller setting $\lambda_\star$, we optimize this parameter to balance the two terms. Similarly, the per-round exploitation regret improves with a larger setting $n_\star$, while the cumulative exploration regret improves with a smaller setting $n_\star$, and our choice of $n_\star$ optimizes this tradeoff.

An important definition that will appear throughout the analysis is the expected reward variance, when a single action is chosen uniformly at random:

$$V := \mathbb{E}_{(x,y)\sim\mathcal{D}}\left[\frac{1}{K}\sum_{a\in\mathcal{A}} y^2(a) - \left(\frac{1}{K}\sum_{a\in\mathcal{A}} y(a)\right)^2\right]. \tag{22}$$

### F.1  Estimating $V$

The first step is a deviation bound for estimating $V$.

**Lemma 16.** *After $n_\star$ rounds, the estimate $\hat{V}$ satisfies, with probability at least $1 - \delta$,*

$$|\hat{V} - V| \leq \sqrt{\frac{V\ln(2/\delta)}{n_\star}} + \frac{\ln(2/\delta)}{6n_\star}.$$

*Proof.* Note that our estimator, $\hat{V} = \frac{1}{n_\star}\sum_{t=1}^{n_\star} Z_t$, is an average of i.i.d. terms, with

$$Z_t := \frac{1}{2K^2}\sum_{a,b\in\mathcal{A}} (y_t(a) - y_t(b))^2 \frac{\mathbf{1}(a,b\in A_t)}{U(a,b\in A_t)},$$

where $U$ is a uniform distribution over all rankings. The mean of this random variable is precisely $V$:

$$\mathbb{E}_{(x,y)\sim\mathcal{D},A\sim U}[Z_t] = \frac{1}{2K^2}\mathbb{E}_{x,y}\left[\sum_{a,b\in\mathcal{A}} (y(a) - y(b))^2\right]$$

$$= \mathbb{E}_{x,y}\left[\frac{1}{2K^2}\sum_{a,b\in\mathcal{A}}\left(y(a)^2 - 2y(a)y(b) + y(b)^2\right)\right]$$

$$= \mathbb{E}_{x,y}\left[\frac{1}{K}\sum_a y(a)^2 - \left(\frac{1}{K}\sum_a y(a)\right)^2\right] = V.$$

Since we choose $L$ actions uniformly at random, the probability for two distinct actions jointly being selected is $U(a,b\in A) = \frac{L(L-1)}{K(K-1)}$ and for a single action it is $U(a\in A) = L/K$. The $(y(a) - y(b))^2$ term is at most one but it is always zero for $a = b$, so the range of $Z_t$ is at most

$$0 \leq Z_t \leq \frac{1}{2K^2}\sum_{a\neq b\in A_t}\frac{K(K-1)}{L(L-1)} = \frac{K(K-1)}{2K^2} \leq \frac{1}{2}.$$

Note that the last summation is only over the $L(L-1)$ action pairs corresponding to the slate $A_t$, as the indicator in $Z_t$ eliminates the other terms in the sum over all actions from $\mathcal{A}$.

As for the second moment, since $Z_t \in [0, 1/2]$, we have

$$\mathbb{E}[Z_t^2] \leq \mathbb{E}[Z_t]/2 \leq V/2.$$

By Bernstein's inequality, we are guaranteed that with probability at least $1 - \delta$, after $n_\star$ rounds,

$$|\hat{V} - V| \leq \sqrt{\frac{V\ln(2/\delta)}{n_\star}} + \frac{\ln(2/\delta)}{6n_\star}. \qquad \square$$

Equipped with the deviation bound we can complete the square to find that

$$V - \sqrt{\frac{V \ln(2/\delta)}{n_\star}} + \frac{\ln(2/\delta)}{4n_\star} \leq \hat{V} + \frac{5\ln(2/\delta)}{12n_\star}$$

$$\Rightarrow \left( \sqrt{V} - \sqrt{\frac{\ln(2/\delta)}{4n_\star}} \right)^2 \leq \hat{V} + \frac{\ln(2/\delta)}{2n_\star}$$

$$\Rightarrow V \leq \left( \sqrt{\frac{\ln(2/\delta)}{4n_\star}} + \sqrt{\hat{V} + \frac{\ln(2/\delta)}{2n_\star}} \right)^2 \leq 2\hat{V} + \frac{3\ln(2/\delta)}{2n_\star}.$$

Our definition of $\lambda_\star$ uses $\tilde{V}$ which is precisely this final upper bound. Working from the other side of the deviation bound, we know that

$$\hat{V} \leq \left( \sqrt{V} + \sqrt{\frac{\ln(2/\delta)}{4n_\star}} \right)^2 \leq 2V + \frac{\ln(2/\delta)}{2n_\star}.$$

And combining the two, we see that

$$V \leq \tilde{V} \leq 4V + \frac{5\ln(2/\delta)}{2n_\star}, \tag{23}$$

with probability at least $1 - \delta$.

### F.2  Parameter Estimation in Linear Regression

To control the regret associated with the exploitation rounds, we also need to bound $\|\hat{w} - w^\star\|_2$ which follows from a standard analysis of linear regression.

At each round $t$, we solve a least squares problem with features $y_t(A_t)$ and response $r_t$ which we know has $\mathbb{E}[r_t \mid y_t, A_t] = y_t(A_t)^T w^\star$. The estimator is

$$w_t := \operatorname*{argmin}_w \sum_{i=1}^t \left( y_i(A_i)^T w - r_i \right)^2.$$

Define the 2nd-moment matrix of reward features,

$$\Sigma_t := \sum_{i=1}^t y_i(A_i) y_i(A_i)^T,$$

which governs the estimation error of the least squares solution as we show in the next lemma.

**Lemma 17.** *Let $\Sigma_t$ denote the 2nd-moment reward matrix after $t$ rounds of interaction and let $w_t$ be the least-squares solution. There is a universal constant $c > 0$ such that for any $\delta \in (0, 2/e)$, with probability at least $1 - \delta$,*

$$\|w_t - w^\star\|_{\Sigma_t}^2 \leq cL \ln(2/\delta).$$

*Proof.* This lemma is the standard analysis of fixed-design linear regression with bounded noise. By definition of the ordinary least squares estimator, we have $\Sigma_t w_t = Y_{1:t}^T r_{1:t}$ where $Y_{1:t} \in \mathbb{R}^{t \times L}$ is the matrix of features, $r_{1:t} \in \mathbb{R}^t$ is the vector of responses and $\Sigma_t = Y_{1:t}^T Y_{1:t}$ is the 2nd-moment matrix of reward features defined above. The true weight vector satisfies $\Sigma_t w^\star = Y_{1:t}^T(r_{1:t} - \xi_{1:t})$ where $\xi_{1:t} \in \mathbb{R}^t$ is the noise. Thus $\Sigma_t(w_t - w^\star) = Y_{1:t}^T \xi_{1:t}$, and therefore,

$$\|w_t - w^\star\|_{\Sigma_t}^2 = (w_t - w^\star)^T \Sigma_t (w_t - w^\star) = (w_t - w^\star)^T \Sigma_t \Sigma_t^\dagger \Sigma_t (w_t - w^\star) = \xi_{1:t}^T Y_{1:t} \Sigma_t^\dagger Y_{1:t}^T \xi_{1:t},$$

where $\Sigma_t^\dagger$ is the pseudoinverse of $\Sigma_t$ and we use the fact that $AA^\dagger A = A$ for any symmetric matrix $A$. Since $\Sigma_t^\dagger = (Y_{1:t}^T Y_{1:t})^\dagger$, the matrix $Y_{1:t} \Sigma_t^\dagger Y_{1:t}^T$ is a projection matrix, and it can be written as $UU^T$ where $U \in \mathbb{R}^{t \times L'}$ is a matrix with $L'$ orthonormal columns where $L' \leq L$. We now have to bound

the term $\|U^T\xi_{1:t}\|_2^2 = \xi_{1:t}^T U U^T \xi_{1:t}$. Let $H_{xy} = (x_1, y_1, \ldots, x_t, y_t)$ denote the history excluding the noise. Conditioned on $H_{xy}$, the vector $\xi_{1:t}$ is a subgaussian random vector with independent components, so we can apply subgaussian tail bounds. Applying Lemma 25, due to Rudelson and Vershynin [25], we see that with probability at least $1 - \delta$,

$$\xi_{1:t}^T U U^T \xi_{1:t} \leq \mathbb{E}[\xi_{1:t}^T U U^T \xi_{1:t} \mid H_{xy}] + \sqrt{c_0 \|UU^T\|_F^2 \ln(2/\delta)} + c_0 \|UU^T\| \ln(2/\delta) \quad (24)$$

$$\leq \left( L + \sqrt{c_0 L \ln(2/\delta)} + c_0 \ln(2/\delta) \right) \leq \left( \sqrt{L} + \sqrt{c_0 \ln(2/\delta)} \right)^2.$$

To derive the second line, we use the fact that $UU^T$ is a projection matrix for an $L'$-dimensional subspace, so its Frobenius norm is bounded as $\|UU^T\|_F^2 = \mathrm{tr}(UU^T) = L' \leq L$, while its spectral norm is $\|UU^T\| = 1$. The expectation in Eq. (24) is bounded using the conditional independence of the noise and the fact that its conditional expectation is zero:

$$\mathbb{E}[\xi_{1:t}^T U U^T \xi_{1:t} \mid H_{xy}] = \mathrm{tr}\Big( U U^T \mathbb{E}[\xi_{1:t}\xi_{1:t}^T \mid H_{xy}]\Big) = \mathrm{tr}\Big( U U^T \mathrm{diag}(\mathbb{E}[\xi_i^2 \mid x_i, y_i])_{i \leq t}\Big)$$

$$\leq \mathrm{tr}(UU^T)\Big( \max_{i \leq t} \mathbb{E}[\xi_i^2 \mid x_i, y_i]\Big) \leq L' \leq L.$$

Finally, when $\delta \in (0, 2/e)$ and with $c = (1 + \sqrt{c_0})^2$, we obtain the desired bound. $\qquad \square$

### F.3  Analysis of the 2nd-Moment Matrix $\Sigma_t$

We now show that the 2nd-moment matrix of reward features has large eigenvalues. This lets us translate the error in Lemma 17 to the Euclidean norm, which will play a role in bounding the exploitation regret. Interestingly, the lower bound on the eigenvalues is related to the exploration regret, so we can explore until the eigenvalues are large, without incurring too much regret.

To prove the bound, we use a full sequence of exploration data, which enables us to bypass the data-dependent stopping time. Let $\{x_t, y_t, A_t, \xi_t\}_{t=1}^T$ be a sequence of random variables where $(x, y, \xi) \sim \mathcal{D}$ and $A_t$ is drawn uniformly at random. Let $w_t$ be the least squares solution on the data in this sequence up to round $t$, and let $\Sigma_t$ be the 2nd-moment matrix of the reward features.

**Lemma 18.** *With probability at least $1 - \delta$, for all $t \leq T$,*

$$\Sigma_t \succeq \Big( tV - 4L\sqrt{tV \ln(4LT/\delta)} - 4L \ln(4LT/\delta)\Big) I_L,$$

*where $I_L$ is the $L \times L$ identity matrix.*

*Proof.* For $K = 1$, we have $V = 0$, so the bound holds. In the remainder, assume $K \geq 2$. The proof has two components: the spectral decomposition of the mean $\mathbb{E}\Sigma_t$ and the deviation bound on $\Sigma_t$.

**Spectral decomposition of $\mathbb{E}\Sigma_t$:** The first step in the proof is to analyze the expected value of the 2nd-moment matrix. Since $y_t, A_t$ are identically distributed, it suffices to consider just one term. Fixing $x$ and $y$, we only reason about the randomness in picking $A$. Let $S := \mathbb{E}_{A \sim U}[y(A)y(A)^T] \in \mathbb{R}^{L \times L}$ be the mean matrix for that round. We have:

$$z^T S z = \sum_{\ell=1}^L z_\ell^2 \sum_{a \in \mathcal{A}} \frac{1}{K} y(a)^2 + \sum_{\ell \neq \ell'} z_\ell z_{\ell'} \sum_{a \neq a' \in \mathcal{A}} \frac{y(a)y(a')}{K(K-1)}$$

$$= \frac{\|y\|_2^2 \|z\|_2^2}{K} + \sum_{\ell \neq \ell'} z_\ell z'_\ell \sum_{a, a' \in \mathcal{A}} \frac{y(a)y(a')}{K(K-1)} - \sum_{\ell \neq \ell'} z_\ell z'_\ell \sum_a \frac{y(a)^2}{K(K-1)}.$$

Define $\bar{y} := \frac{1}{K} \sum_{a \in \mathcal{A}} y(a)$, $E_y^2 := \frac{1}{K} \sum_{a \in \mathcal{A}} y(a)^2$, and $V_y := E_y^2 - \bar{y}^2$, and observe that by the definition of $V$ in Eq. (22), we have $\mathbb{E}_{x,y} V_y = V$. Continuing the derivation, we obtain:

$$z^T S z = E_y^2 \|z\|_2^2 + \sum_{\ell \neq \ell'} z_\ell z_{\ell'} \left( \frac{K}{K-1}\bar{y}^2 - \frac{1}{K-1}E_y^2 \right)$$

$$= E_y^2 \|z\|_2^2 + \Big((z^T \mathbf{1})^2 - \|z\|_2^2\Big)\left( \frac{K}{K-1}\bar{y}^2 - \frac{1}{K-1}E_y^2 \right)$$

$$= \frac{K}{K-1}V_y \|z\|_2^2 + (z^T \mathbf{1})^2 \left( \frac{K}{K-1}\bar{y}^2 - \frac{1}{K-1}E_y^2 \right).$$

To finish the derivation, let $u = \mathbf{1}/\sqrt{L}$ be the unit vector in the direction of all ones and $P = I - uu^T$ be the projection matrix on the subspace orthogonal with $u$. Then

$$z^T S z = \frac{K}{K-1} V_y (z^T u u^T z + z^T P z) + L(z^T u u^T z)\left(\bar{y}^2 - \frac{1}{K-1} V_y\right)$$

$$= s\left(\frac{K-L}{K-1} V_y + L\bar{y}^2\right)(z^T u u^T z) + \frac{K}{K-1} V_y(z^T P z).$$

Thus,

$$S = \left(\frac{K-L}{K-1} V_y + L\bar{y}^2\right) uu^T + \frac{K}{K-1} V_y P.$$

By taking the expectation, we obtain the spectral decomposition with eigenvalues $\lambda_u$ and $\lambda_P$ associated, respectively, with $uu^T$ and $P$:

$$\mathbb{E}_{x,y,A}[y(A)y(A)^T] = \mathbb{E}_{x,y}[S] = \underbrace{\left(\frac{K-L}{K-1} V + L\mathbb{E}[\bar{y}^2]\right)}_{\lambda_u} uu^T + \underbrace{\left(\frac{K}{K-1} V\right)}_{\lambda_P} P. \quad (25)$$

We next bound the eigenvalue $\lambda_u$. By positivity of $y$, note that $E_y^2 \leq (\max_a y(a))\bar{y} \leq K\bar{y}^2$. Therefore, $V_y = E_y^2 - \bar{y}^2 \leq (K-1)\bar{y}^2$, and thus $\mathbb{E}[\bar{y}^2] \geq V/(K-1)$, so

$$\lambda_u = \frac{K-L}{K-1} V + L\mathbb{E}[\bar{y}^2] \geq \frac{K}{K-1} V.$$

Thus, both eigenvalues are lower bounded by $\frac{K}{K-1} V \geq V$.

**The deviation bound:** For deviation bound, we follow the spectral structure of $\mathbb{E}\Sigma_t$ and first reason about the properties of $\Sigma_t u$, followed by the analysis of $P\Sigma_t P$. Throughout the analysis, let $z_i := y_i(A_i)$ denote the $L$-dimensional reward feature vector on round $i$, and consider a fixed $t \leq T$.

**Direction $u$:** We begin by the analysis of $\Sigma_t u$. Specifically, we will show that $\|\Sigma_t u - (\mathbb{E}\Sigma_t)u\|_2$ is small. We apply Bernstein's inequality to a single coordinate $\ell$, then take a union bound to obtain a bound on $\|\cdot\|_\infty$, and convert to a bound on $\|\cdot\|_2$. For a fixed $\ell$ and $i \leq t$, define

$$X_i := z_{i\ell} z_i^T u$$

and note that $(\Sigma_t u)_\ell = \sum_{i \leq t} X_i$. The range and variance of $X_i$ are bounded as

$$0 \leq X_i \leq \sqrt{L}$$
$$\mathbb{E}[X_i^2] = \mathbb{E}[z_{i\ell}^2 (z_i^T u)^2] \leq \mathbb{E}[(z_i^T u)^2] = u^T \mathbb{E}[z_i z_i^T] u = \lambda_u$$

where the last equality follows by Eq. (25). Thus, by Bernstein's inequality, with probability at least $1 - \delta/2L$,

$$\left|\sum_{i \leq t} X_i - \sum_{i \leq t} \mathbb{E}X_i\right| \leq \sqrt{2t\lambda_u \ln(4L/\delta)} + \sqrt{L}\ln(4L/\delta)/3.$$

Taking a union bound over $\ell \leq L$ yields that with probability at least $1 - \delta/2$,

$$\|\Sigma_t u - (\mathbb{E}\Sigma_t)u\|_2 \leq \sqrt{L}\|\Sigma_t u - (\mathbb{E}\Sigma_t)u\|_\infty \leq \sqrt{2Lt\lambda_u \ln(4L/\delta)} + L\ln(4L/\delta)/3. \quad (26)$$

**Orthogonal to $u$:** In the subspace orthogonal to $u$, we apply the matrix Bernstein inequality. Let $X_i$, for $i \leq t$, be the matrix random variable

$$X_i := Pz_i z_i^T P - P\mathbb{E}[z_i z_i^T]P$$

and note that $\sum_{i \leq t} X_i = P\Sigma_t P - \mathbb{E}[P\Sigma_t P]$. Since $z_i$ are i.i.d., below we analyze a single $z_i$ and $X_i$ and drop the index $i$. The range can be bounded as

$$\lambda_{\max}(X) \leq \lambda_{\max}(Pzz^T P) \leq \|z\|_2^2 \leq L.$$

To bound the variance, we use Schatten norms, i.e., $L_p$ norms applied to the spectrum of a symmetric matrix. The Schatten $p$-norm is denoted as $\|\cdot\|_{\sigma,p}$. Note that the operator norm is $\|\cdot\|_{\sigma,\infty}$ and the

trace norm is $\|\cdot\|_{\sigma,1}$. We begin by upper-bounding the variance by the second moment, then use the convexity of the norm, the monotonicity of Schatten norms, and the fact that the trace norm of a positive semi-definite matrix equals its trace to obtain:

$$\left\|\mathbb{E}[X^2]\right\|_{\sigma,\infty} \leq \left\|\mathbb{E}[(Pzz^TP)^2]\right\|_{\sigma,\infty}$$

$$\leq \mathbb{E}\left[\|(Pzz^TP)^2\|_{\sigma,\infty}\right] \leq \mathbb{E}\left[\|(Pzz^TP)^2\|_{\sigma,1}\right] = \mathbb{E}\left[\mathrm{tr}\left((Pzz^TP)^2\right)\right],$$

and continue by the matrix Holder inequality, $\mathrm{tr}(A^TB) \leq \|A\|_{\sigma,\infty}\|B\|_{\sigma,1}$, and Eq. (25) to obtain:

$$\leq \left(\max_z \|Pzz^TP\|_{\sigma,\infty}\right)\mathbb{E}\left[\|Pzz^TP\|_{\sigma,1}\right]$$

$$\leq L \,\mathrm{tr}\,\mathbb{E}\left[Pzz^TP\right] = L(L-1)\lambda_P.$$

Reverting to the notation $\|\cdot\|$ for the operator norm, the matrix Bernstein inequality (Lemma 26) yields that with probability at least $1 - \delta/2$,

$$\left\|P\Sigma_tP - \mathbb{E}[P\Sigma_tP]\right\| = \left\|\sum_{i\leq t}X_i - \sum_{i\leq t}\mathbb{E}X_i\right\| \leq \sqrt{2L^2t\lambda_P\ln(2L/\delta)} + 2L\ln(2L/\delta)/3. \quad (27)$$

**The final bound:** Let $x$ be an arbitrary unit vector. Decompose it along the all-ones direction and the orthogonal direction as $x = \alpha u + \beta v$, where $v \perp u$, and $\alpha^2 + \beta^2 = 1$. Let $v' = (\Sigma_t - \mathbb{E}\Sigma_t)u$. Then

$$\left|x^T(\Sigma_t - \mathbb{E}\Sigma_t)x\right| = \left|\alpha u^T(\Sigma_t - \mathbb{E}\Sigma_t)x + \beta v^T(\Sigma_t - \mathbb{E}\Sigma_t)\alpha u + \beta v^T(\Sigma_t - \mathbb{E}\Sigma_t)\beta v\right|$$

$$\leq |\alpha| \cdot \left\|u^T(\Sigma_t - \mathbb{E}\Sigma_t)\right\|_2 + |\alpha\beta| \cdot \left\|(\Sigma_t - \mathbb{E}\Sigma_t)u\right\|_2 + \beta^2\left|v^T(\Sigma_t - \mathbb{E}\Sigma_t)v\right|$$

$$\leq 2|\alpha| \cdot \|v'\|_2 + |\beta| \cdot \|P\Sigma_tP - \mathbb{E}[P\Sigma_tP]\| \ . \quad (28)$$

From Eq. (25), we have

$$x^T(\mathbb{E}\Sigma_t)x \geq \alpha^2t\lambda_u + \beta^2t\lambda_P. \quad (29)$$

To finish the proof, we will use the identity valid for all $A, B, c \geq 0$

$$A + B - c\sqrt{A+B} \geq A + B - c\sqrt{A} - c\sqrt{B} + c^2/4 - c^2/4$$

$$= A - c\sqrt{A} + \left(\sqrt{B} - c/2\right)^2 - c^2/4$$

$$\geq A - c\sqrt{A} - c^2/4. \quad (30)$$

Combining Eq. (28) and Eq. (29), and plugging in bounds from Eq. (26) and Eq. (27), we have

$$x^T\Sigma_tx \geq \alpha^2t\lambda_u + \beta^2t\lambda_P - 2|\alpha| \cdot \|v'\|_2 - |\beta| \cdot \|P\Sigma_tP - \mathbb{E}[P\Sigma_tP]\|$$

$$\geq \alpha^2t\lambda_u + \beta^2t\lambda_P - 2|\alpha|\sqrt{2Lt\lambda_u\ln(4L/\delta)} - \frac{2|\alpha|L}{3}\ln(4L/\delta)$$

$$- |\beta|\sqrt{2L^2t\lambda_P\ln(2L/\delta)} - \frac{2|\beta|L}{3}\ln(2L/\delta)$$

$$\geq \alpha^2t\lambda_u + \beta^2tV - \left(2\sqrt{2L\ln(4L/\delta)}\right)\sqrt{\alpha^2t\lambda_u}$$

$$- |\beta|\sqrt{4L^2tV\ln(4L/\delta)} - 2L\ln(4L/\delta),$$

where we used $V \leq \lambda_P \leq 2V$, and $|\alpha| \leq 1, |\beta| \leq 1$. We now apply Eq. (30) with $A + B = \alpha^2t\lambda_u$ and $A = \alpha^2tV$ to obtain

$$x^T\Sigma_tx \geq \alpha^2tV + \beta^2tV - \left(2\sqrt{2L\ln(4L/\delta)}\right)\sqrt{\alpha^2tV} - 2L\ln(4L/\delta)$$

$$- 2L|\beta|\sqrt{tV\ln(4L/\delta)} - 2L\ln(4L/\delta)$$

$$\geq tV - 2L\sqrt{2}\left(|\alpha| + |\beta|\right)\sqrt{tV\ln(4L/\delta)} - 4L\ln(4L/\delta)$$

$$\geq tV - 4L\sqrt{tV\ln(4L/\delta)} - 4L\ln(4L/\delta),$$

where we used $|\alpha| + |\beta| \leq \sqrt{2\alpha^2 + 2\beta^2} = \sqrt{2}$. The lemma follows by the union bound over $t \leq T$. $\qquad\square$

### F.4 Analysis of the Exploration Regret

The analysis here is made complicated by the fact that the stopping time of the exploration phase is a random variable. If we let $\hat{t}$ denote the last round of the exploration phase, this quantity is a random variable that depends on the history of interaction up to and including round $\hat{t}$. Our proof here will use a non-random bound $t^\star$ that satisfies $\mathbb{P}(\hat{t} \le t^\star) \ge 1 - \delta$. We will compute $t^\star$ based on our analysis of the 2nd-moment matrix $\Sigma_t$.

A trivial bound on the exploration regret is

$$\sum_{t=1}^{t^\star} \Big[ r_t(\pi^\star(x_t)) - r_t(A_t) \Big] \le t^\star \|w^\star\|_2 \sqrt{L}, \tag{31}$$

which follows from the Cauchy-Schwarz inequality and the fact that the reward features are in $[0,1]$.

In addition, we also bound the exploration regret by the following more precise bound:

**Lemma 19** (Exploration Regret Lemma). *Let $t^\star$ be a non-random upper bound on the random variable $\hat{t}$ satisfying $\mathbb{P}(\hat{t} \le t^\star) \ge 1 - \delta$. Then with probability at least $1 - 2\delta$, the exploration regret is*

$$\sum_{t=1}^{\hat{t}} \Big[ r_t(\pi^\star(x_t)) - r_t(A_t) \Big] \le t^\star \|w^\star\|_2 \min\left\{ \sqrt{KV}, \sqrt{L} \right\} + \|w^\star\|_2 \sqrt{2Lt^\star \ln(1/\delta)}.$$

*Proof.* Let $\{x_t, y_t, A_t, \xi_t\}_{t=1}^{T}$ be a sequence of random variables where $(x, y, \xi) \sim \mathcal{D}$ and $A_t$ is drawn uniformly at random. We are interested in bounding the probability of the event

$$\mathcal{E} := \left\{ \sum_{t=1}^{\hat{t}} \Big( y_t(\pi^\star(x_t)) - y_t(A_t) \Big)^T w^\star \le \epsilon \right\}.$$

This term is exactly the exploration regret, so we want to make sure the probability of this event is large. We first apply the upper bound

$$\sum_{t=1}^{\hat{t}} \Big( y_t(\pi^\star(x_t)) - y_t(A_t) \Big)^T w^\star \le \sum_{t=1}^{\hat{t}} \Big( y_t(A_t^\star) - y_t(A_t) \Big)^T w^\star,$$

where $A_t^\star = \arg\max_A y_t(A)^T w^\star$ is the best possible ranking. This upper bound ensures that every term in the sum is non-negative. We next remove the dependence on the random stopping time $\hat{t}$ and replace it with a deterministic number of terms $t^\star$:

$$\mathbb{P}(\mathcal{E}) \ge \mathbb{P}\left( \sum_{t=1}^{\hat{t}} \Big( y_t(A_t^\star) - y_t(A_t) \Big)^T w^\star \le \epsilon \right)$$

$$\ge \mathbb{P}\left( \sum_{t=1}^{\hat{t}} \Big( y_t(A_t^\star) - y_t(A_t) \Big)^T w^\star \le \epsilon \cap \hat{t} \le t^\star \right)$$

$$\ge \mathbb{P}\left( \sum_{t=1}^{t^\star} \Big( y_t(A_t^\star) - y_t(A_t) \Big)^T w^\star \le \epsilon \cap \hat{t} \le t^\star \right)$$

$$\ge 1 - \mathbb{P}\left( \sum_{t=1}^{t^\star} \Big( y_t(A_t^\star) - y_t(A_t) \Big)^T w^\star > \epsilon \right) - \mathbb{P}\left( \hat{t} > t^\star \right)$$

$$\ge 1 - \delta - \mathbb{P}\left( \sum_{t=1}^{t^\star} \Big( y_t(A_t^\star) - y_t(A_t) \Big)^T w^\star > \epsilon \right).$$

The first line follows from the definition of $A_t^\star$ which only increases the sum, so decreases the probability of the event. The second inequality is immediate, while the third inequality holds because

all terms of the sequence are non-negative. The fourth inequality is the union bound and the last is by assumption on the event $\{\hat{t} \leq t^\star\}$.

Now we can apply a standard concentration analysis. The mean of the random variables is

$$
\begin{aligned}
\mathbb{E}_{x,y,A}\Big[\big(y(A^\star) - y(A)\big)^T w^\star\Big] &\leq \|w^\star\|_2 \Big\|\mathbb{E}_{x,y,A}\big[y(A^\star) - y(A)\big]\Big\|_2 \\
&= \|w^\star\|_2 \sqrt{\sum_{\ell \leq L} \mathbb{E}_{x,y}\left[y(A_\ell^\star) - \bar{y}\right]^2} \\
&\leq \|w^\star\|_2 \sqrt{\sum_{\ell \leq L} \mathbb{E}_{x,y}\left[\big(y(A_\ell^\star) - \bar{y}\big)^2\right]} \\
&\leq \|w^\star\|_2 \sqrt{K} \sqrt{\frac{1}{K}\sum_{a \in \mathcal{A}} \mathbb{E}_{x,y}\left[\big(y(a) - \bar{y}\big)^2\right]} \\
&= \|w^\star\|_2 \sqrt{KV}.
\end{aligned}
$$

The first inequality is Cauchy-Schwarz while the second is Jensen's inequality and the third comes from adding non-negative terms. The range of the random variable is bounded as

$$
\sup_{x,y,A}\left|\big(y(A^\star) - y(A)\big)^T w^\star - \mathbb{E}_{x,y,A}\Big[\big(y(A^\star) - y(A)\big)^T w^\star\Big]\right| \leq \|w^\star\|_2 \sqrt{L},
$$

because $0 \leq \big(y(A^\star) - y(A)\big)^T w^\star \leq \|w^\star\|_2 \sqrt{L}$. Thus by Hoeffding's inequality, with probability at least $1 - \delta$,

$$
\begin{aligned}
\sum_{t=1}^{t^\star}\big(y(A^\star) - y(A)\big)^T w^\star &\leq \sum_{t=1}^{t^\star}\mathbb{E}_{x,y,A}\Big[\big(y(A^\star) - y(A)\big)^T w^\star\Big] + \|w^\star\|_2 \sqrt{2Lt^\star \ln(1/\delta)} \\
&\leq t^\star \|w^\star\|_2 \sqrt{KV} + \|w^\star\|_2 \sqrt{2Lt^\star \ln(1/\delta)}.
\end{aligned}
$$

Combining this bound with the bound of Eq. (31) proves the lemma. $\qquad\square$

### F.5 Analysis of the Exploitation Regret

In this section we show that after the exploration rounds, we can find a policy that has low expected regret. The technical bulk of this section involves a series of deviation bounds showing that we have good estimates of the expected reward for each policy.

In addition to $\bar{y}$ from the previous sections, we will also need the sample quantity $\bar{y}_t := \frac{1}{K}\sum_{a \in \mathcal{A}} y_t(a)$, which will allow us to relate the exploitation regret to the variance term $V$. Since we are using uniform exploration, the importance-weighted feature vectors are as follows:

$$
\hat{y}_t(a) = \frac{\mathbf{1}(a \in A_t) y_t(a)}{U(a \in A_t)} = \frac{K}{L}\mathbf{1}(a \in A_t) y_t(a).
$$

Given any estimate $\hat{w}$ of the true weight vector $w^\star$, the empirical reward estimate for a policy $\pi$ is

$$
\eta_n(\pi, \hat{w}) := \frac{1}{n}\sum_{t=1}^{n} \hat{y}_t(\pi(x_t))^T \hat{w}.
$$

A natural way to show that the policy with a low empirical reward has also a low expected regret is to show that for all policies $\pi$, the empirical reward estimate $\eta_n(\pi, \hat{w})$ is close to the true reward, $\eta(\pi)$, defined as,

$$
\eta(\pi) := \mathbb{E}_{x,y}\left[y(\pi(x))^T w^\star\right].
$$

Rather than bounding the deviation of $\eta_n$ directly, we instead control a shifted version of $\eta_n$, namely,

$$
\psi_n(\pi, \hat{w}) := \frac{1}{n}\sum_{t=1}^{n}\left[\hat{y}_t(\pi(x_t))^T \hat{w} - \bar{y}_t \mathbf{1}^T \hat{w}\right],
$$

where $\mathbf{1}$ is the $L$-dimensional all-ones vector. Note that $\bar{y}_t$ is based on the rewards of all actions, even those that were not chosen at round $t$. This is not an issue, since $\bar{y}_t$ is only used in the analysis.

**Lemma 20.** *Fix $\delta \in (0,1)$ and assume that $\|\hat{w} - w^\star\|_2 \le \theta$ for some $\theta \ge 0$. For any $\delta \in (0,1)$, with probability at least $1 - \delta$, we have that for all $\pi \in \Pi$,*

$$\left| \psi_n(\pi, \hat{w}) - \eta(\pi, w^\star) + \mathbb{E}_{x,y}[\bar{y}\mathbf{1}^T w^\star] \right|$$

$$\le 2(\theta + \|w^\star\|_2)\sqrt{K}\left( \sqrt{\frac{\ln(2N/\delta)}{n}} + \sqrt{\frac{K}{L}\frac{\ln(2N/\delta)}{n}} \right) + \theta \min\{\sqrt{KV}, 2\sqrt{L}\}.$$

*Proof.* We add and subtract several terms to obtain a decomposition. We introduce the shorthands $y_\pi := y(\pi(x))$, $\hat{y}_\pi := \hat{y}(\pi(x))$, and $\hat{y}_{t,\pi} := \hat{y}_t(\pi(x_t))$.

$$\psi_n(\pi, \hat{w}) - \eta(\pi, w^\star) + \mathbb{E}_{x,y}[\bar{y}\mathbf{1}^T w^\star]$$

$$= \frac{1}{n}\sum_{t=1}^n (\hat{y}_{t,\pi} - \bar{y}_t\mathbf{1})^T \hat{w} - \mathbb{E}_{x,y}[y_\pi - \bar{y}\mathbf{1}]^T w^\star$$

$$= \underbrace{\frac{1}{n}\sum_{t=1}^n \left( (\hat{y}_{t,\pi} - \bar{y}_t\mathbf{1})^T \hat{w} - \mathbb{E}_{x,y}[y_\pi - \bar{y}\mathbf{1}]^T \hat{w} \right)}_{\text{Term 1}} + \underbrace{\mathbb{E}_{x,y}[y_\pi - \bar{y}\mathbf{1}]^T(\hat{w} - w^\star)}_{\text{Term 2}}.$$

There are two terms to bound here. We bound the first term by Bernstein's inequality, using that fact that $\hat{y}_t$ is coordinate-wise unbiased for $y$. The second term will be bounded via a deterministic analysis, which will yield an upper bound related to the reward-feature variance $V$.

**Term 1:** Note that each term of the sum has expectation zero, since $\hat{y}_t$ is an unbiased estimate. Moreover, the range of each individual term in the sum can be bounded as

$$\left| (\hat{y}_{t,\pi} - \bar{y}_t\mathbf{1} - \mathbb{E}_{x,y}[y_\pi - \bar{y}\mathbf{1}])^T \hat{w} \right| \le \|\hat{w}\|_2 \left\| \hat{y}_{t,\pi} - \bar{y}_t\mathbf{1} - \mathbb{E}_{x,y}[y_\pi - \bar{y}\mathbf{1}] \right\|_2$$

$$\le (\theta + \|w^\star\|_2)\frac{2K}{\sqrt{L}}.$$

The second line is derived by bounding the two factors separately. The first factor is bounded by the triangle inequality: $\|\hat{w}\|_2 \le \|w^\star\|_2 + \|\hat{w} - w^\star\|_2 \le \|w^\star\|_2 + \theta$. The second factor is a norm of an $L$-dimensional vector. The vector $\hat{y}_{t,\pi}$ has coordinates in $[0, K/L]$, whereas the coordinates of $\bar{y}_t\mathbf{1}$, $y_\pi$, and $\bar{y}\mathbf{1}$ are all in $[0,1]$, so the final vector has coordinates in $[-2, K/L + 1]$, and its Euclidean norm is thus at most $\sqrt{L}(2K/L)$ since $K \ge L$.

The variance can be bounded by the second moment, which is

$$\mathbb{E}_{x,y,A}\left[ ((\hat{y}_\pi - \bar{y}\mathbf{1})^T \hat{w})^2 \right] \le \|\hat{w}\|_2^2 \mathbb{E}_{x,y,A}\left[ \sum_{\ell=1}^L (\hat{y}(\pi(x)_\ell) - \bar{y})^2 \right]$$

$$\le \|\hat{w}\|_2^2 \mathbb{E}_{x,y,A}\left[ \sum_{\ell=1}^L \left( \hat{y}(\pi(x)_\ell)^2 + \bar{y}^2 \right) \right]$$

$$\le \|\hat{w}\|_2^2 \sum_{\ell=1}^L \left( \frac{K}{L}\mathbb{E}_{x,y,A}[\hat{y}(\pi(x)_\ell)] + 1 \right)$$

$$= \|\hat{w}\|_2^2 \sum_{\ell=1}^L \left( \frac{K}{L}\mathbb{E}_{x,y}[y(\pi(x)_\ell)] + 1 \right)$$

$$\le 2(\theta + \|w^\star\|_2)^2 K,$$

where the last inequality uses $K \ge L$. Bernstein's inequality implies that with probability at least $1 - \delta$, for all $\pi \in \Pi$,

$$\left| \frac{1}{n}\sum_{t=1}^n \left( (\hat{y}_{t,\pi} - \bar{y}_t\mathbf{1})^T \hat{w} - \mathbb{E}_{x,y}[y_\pi - \bar{y}\mathbf{1}]^T \hat{w} \right) \right| \le (\theta + \|w^\star\|_2)\left[ \sqrt{\frac{4K\ln(2N/\delta)}{n}} + \frac{4K\ln(2N/\delta)}{3n\sqrt{L}} \right].$$

**Term 2:** For the second term, we use the Cauchy-Schwarz inequality,

$$\mathbb{E}_{x,y}[y_\pi - \bar{y}\mathbf{1}]^T(\hat{w} - w^\star) \le \left\| \mathbb{E}_{x,y}[y_\pi - \bar{y}\mathbf{1}] \right\|_2 \|\hat{w} - w^\star\|_2$$

The difference in the weight vectors will be controlled by our analysis of the least squares problem. We need to bound the other quantity here and we will use two different bounds. First,

$$\left\|\mathbb{E}_{x,y}[y_\pi - \bar{y}\mathbf{1}]\right\|_2 \le \mathbb{E}_{x,y}\|y_\pi - \bar{y}\mathbf{1}\|_2 \le \mathbb{E}_{x,y}\|y_\pi\|_2 + \bar{y}\|\mathbf{1}\|_2 \le 2\sqrt{L}.$$

Second,

$$\|\mathbb{E}_{x,y}[y_\pi - \bar{y}\mathbf{1}]\|_2 = \sqrt{\mathbb{E}_{x,y}\sum_{\ell=1}^{L}\big(y(\pi(x)_\ell) - \bar{y}\big)^2} \le \sqrt{\mathbb{E}_{x,y}\sum_{a\in\mathcal{A}}(y(a) - \bar{y})^2}$$

$$= \sqrt{K}\sqrt{\mathbb{E}_{x,y}\frac{1}{K}\sum_{a\in\mathcal{A}}(y(a) - \bar{y})^2} = \sqrt{KV}.$$

**Combining everything:** Putting everything together, we obtain the bound

$$(\theta + \|w^\star\|_2)\left[\sqrt{\frac{4K\ln(2N/\delta)}{n}} + \frac{4K\ln(2N/\delta)}{3n\sqrt{L}}\right] + \theta\min\{\sqrt{KV}, 2\sqrt{L}\}.$$

Collecting terms together proves the main result. $\qquad\square$

Assume that we explore for $\hat{t}$ rounds and then call AMO with weight vector $\hat{w}$ and importance-weighted rewards $\hat{y}_1,\dots\hat{y}_{\hat{t}}$ to produce a policy $\hat{\pi}$ that maximizes $\eta_{\hat{t}}(\pi, \hat{w})$. In the remaining exploitation rounds we act according to $\hat{\pi}$. With an application of Lemma 20, we can then bound the regret in the exploitation phase. Note that the algorithm ensures that $\hat{t}$ is at least equal to the deterministic quantity $n_\star$, so we can remove the dependence on the random variable $\hat{t}$:

**Lemma 21** (Exploitation Regret Lemma). *Assume that we explore for $\hat{t}$ rounds, where $\hat{t} \ge n_\star$, and we find $\hat{w}$ satisfying $\|\hat{w} - w^\star\|_2 \le \theta$. Then for any $\delta \in (0, 1)$, with probability at least $1 - 2\delta$, the exploitation regret is at most*

$$\sum_{t=\hat{t}+1}^{T}\left[r_t(\pi^\star(x_t)) - r_t(\hat{\pi}(x_t))\right] \le 4T(\theta + \|w^\star\|_2)\sqrt{K}\left(\sqrt{\frac{\ln(2N/\delta)}{n_\star}} + \sqrt{\frac{K}{L}\frac{\ln(2N/\delta)}{n_\star}}\right)$$

$$+ 2T\theta\min\{\sqrt{KV}, 2\sqrt{L}\} + \|w^\star\|_2\sqrt{2LT\ln(1/\delta)}.$$

*Proof.* Using Lemma 20 and the optimality of $\hat{w}$ for the importance-weighted rewards, with probability at least $1 - \delta$, the expected per-round regret of $\hat{\pi}$ is at most

$$\eta(\pi^\star, w^\star) - \eta(\hat{\pi}, w^\star)$$
$$= \left[\eta(\pi^\star, w^\star) - \psi_{\hat{t}}(\pi^\star, \hat{w})\right] + \left[\psi_{\hat{t}}(\hat{\pi}, \hat{w}) - \eta(\hat{\pi}, w^\star)\right] + \left[\eta_{\hat{t}}(\pi^\star, \hat{w}) - \eta_{\hat{t}}(\hat{\pi}, \hat{w})\right]$$
$$\le 4(\theta + \|w^\star\|_2)\sqrt{K}\left(\sqrt{\frac{\ln(2N/\delta)}{\hat{t}}} + \sqrt{\frac{K}{L}\frac{\ln(2N/\delta)}{\hat{t}}}\right) + 2\theta\min\{\sqrt{KV}, 2\sqrt{L}\}.$$

To bound the actual exploitation regret, we use Hoeffding's inequality together with the fact that the absolute value of the per-round regret is at most $\|w^\star\|_2\sqrt{L}$, and finally apply bounds $1/\sqrt{\hat{t}} \le 1/\sqrt{n_\star}$ and $1/\hat{t} \le 1/n_\star$ to prove the lemma. $\qquad\square$

### F.6 Proving the Final Bound

The final bound will follow from regret bounds of Lemmas 19 and 21. These bounds depend on parameters $t^\star$, $n_\star$ and $\theta$. The parameter $n_\star$ is specified directly by the algorithm and is assured to be a lower bound on the stopping time. The parameter $t^\star$ needs to be selected to upper-bound the stopping time $\hat{t}$, and $\theta$ to upper-bound $\|\hat{w} - w^\star\|_2$.

**The stopping time bound $t^\star$ and error bound $\theta$:** Our algorithm uses the constants

$$\lambda_\star := \max\left\{6L^2\ln(4LT/\delta), (T\tilde{V}/B)^{2/3}(L\ln(2/\delta))^{1/3}\right\},$$
$$n_\star := T^{2/3}(K\ln(N/\delta))^{1/3}\max\{L^{-1/3}, (BL)^{-2/3}\},$$

and we will show we can set
$$t^\star := \max\left\{6\lambda_\star/V,\, n_\star\right\}, \qquad \theta := \sqrt{cL\ln(2/\delta)/\lambda_\star},$$
where $c$ is the constant from Lemma 17.

Recall that we assume $T \geq (K\ln(N/\delta)/L)\max\{1, (B\sqrt{L})^{-2}\}$, which ensures that $T \geq n_\star$, and that the algorithm stops exploration with the first round $\hat{t}$ such that $\hat{t} \geq n_\star$ and $\lambda_{\min}(\Sigma_{\hat{t}}) > \lambda_\star$. Thus, by Lemma 17, $\theta$ is indeed an upper bound on $\|\hat{w} - w^\star\|_2$. Furthermore, since $t^\star \geq n_\star$, it suffices to argue that $\Sigma_{t^\star} \succeq \lambda_\star I_L$ with probability at least $1 - \delta$. We will show this through Lemma 18.

Specifically, Lemma 18 ensures that after $t^\star$ rounds the 2nd-moment matrix satisfies, with probability at least $1 - \delta$,
$$\Sigma_{t^\star} \succeq \left(t^\star V - 4L\sqrt{t^\star V \ln(4LT/\delta)} - 4L\ln(4LT/\delta)\right) I_L.$$
It suffices to verify that the expression in the parentheses is greater than $\lambda_\star$:
$$\left(t^\star V - 4L\sqrt{t^\star V \ln(4LT/\delta)} - 4L\ln(4LT/\delta)\right) \geq \lambda_\star$$
$$\Leftarrow \quad \left(\sqrt{t^\star V} - 2L\sqrt{\ln(4LT/\delta)}\right)^2 - 4L^2\ln(4LT/\delta) - 4L\ln(4LT/\delta) \geq \lambda_\star$$
$$\Leftarrow \quad \left(\sqrt{t^\star V} - 2L\sqrt{\ln(4LT/\delta)}\right)^2 \geq \lambda_\star + 8L^2\ln(4LT/\delta)$$
$$\Leftarrow \quad \sqrt{t^\star V} - 2L\sqrt{\ln(4LT/\delta)} \geq \sqrt{\lambda_\star + 8L^2\ln(4LT/\delta)}$$
$$\Leftarrow \quad t^\star \geq \frac{1}{V}\left(\sqrt{\lambda_\star + 8L^2\ln(4LT/\delta)} + 2L\sqrt{\ln(4LT/\delta)}\right)^2$$
Our setting is an upper bound on this quantity, using the inequality $(a+b)^2 \leq 2a^2 + 2b^2$ and the fact that $\lambda_\star \geq 6L^2\ln(4LT/\delta)$.

**Regret decomposition:** We next use Lemmas 19 and 21 with the specific values of $t^\star$, $n_\star$ and $\theta$. The leading term in our final regret bound will be on the order $T^{2/3}$. In the smaller-order terms, we ignore polynomial dependence on parameters other than $T$ (such as $K$ and $L$), which we make explicit via $O_T$ notation, e.g., $O(\sqrt{LT}) = O_T(\sqrt{T})$.

The exploration regret is bounded by Lemma 19, using the bound $t^\star \leq 6\lambda_\star/V + n_\star$, and the fact that the exploration vacuously stops at round $T$, so $t^\star$ can be replaced by $\min\{t^\star, T\}$:
$$\text{Exploration Regret} \leq \min\{t^\star, T\}\|w^\star\|_2 \min\{\sqrt{KV}, \sqrt{L}\} + \|w^\star\|_2\sqrt{2LT\ln(1/\delta)}$$
$$\leq \underbrace{\min\left\{\frac{6\lambda_\star}{V}, T\right\} B\min\{\sqrt{KV}, \sqrt{L}\}}_{\text{Term 1}} + \underbrace{n_\star B\sqrt{L}}_{\text{Term 2}} + O_T(\sqrt{T}).$$

Meanwhile, for the exploitation regret, using the fact that $n_\star = \Omega(T^{2/3})$, Lemma 21 yields
$$\text{Exploitation Regret} \leq 4T(\theta + \|w^\star\|_2)\sqrt{K}\left(\sqrt{\frac{\ln(2N/\delta)}{n_\star}} + \sqrt{\frac{K}{L}\frac{\ln(2N/\delta)}{n_\star}}\right)$$
$$+ 2T\theta\min\{\sqrt{KV}, 2\sqrt{L}\} + \|w^\star\|_2\sqrt{2LT\ln(1/\delta)}$$
$$= 4T(\theta + \|w^\star\|_2)\sqrt{\frac{K\ln(2N/\delta)}{n_\star}} + 2T\theta\min\{\sqrt{KV}, 2\sqrt{L}\} + O_T(\sqrt{T})$$
$$\leq \underbrace{4T\left(\sqrt{\frac{cL\ln(2/\delta)}{\lambda_\star}} + B\right)\sqrt{\frac{K\ln(2N/\delta)}{n_\star}}}_{\text{Term 3}}$$
$$+ \underbrace{2T\sqrt{\frac{cL\ln(2/\delta)}{\lambda_\star}}\min\{\sqrt{KV}, 2\sqrt{L}\}}_{\text{Term 4}} + O_T(\sqrt{T}).$$

We now use our settings of $n_\star$ and $\lambda_\star$ to bound all the terms. Working with $\lambda_\star$ is a bit delicate, because it relies on the estimate $\tilde{V}$ rather than $V$. However, by Lemma 16 and Eq. (23), we know that

$$V \leq \tilde{V} \leq 4V + \tau,$$

where $\tau := 5\ln(2/\delta)/(2n_\star)$.

**Term 1:** We proceed by case analysis. First assume that $V \leq \tau$. Then

$$\text{Term } 1 \leq TB\sqrt{KV} \leq TB\sqrt{\frac{5K\ln(2/\delta)}{2n_\star}} \leq \text{Term } 3,$$

so we can use the bound on Term 3 to control this case.

Next assume that $V \geq \tau$, which implies $\tilde{V} \leq 5V$, and distinguish two sub-cases. First, assume that $\lambda_\star$ is the second term in its definition, i.e., $\lambda_\star = (T\tilde{V}/B)^{2/3}(L\ln(2/\delta))^{1/3}$. Then:

$$\begin{aligned}
\text{Term } 1 &\leq \frac{6\lambda_\star}{V} B \min\left\{\sqrt{KV}, \sqrt{L}\right\} \\
&\leq \frac{6B(T\tilde{V}/B)^{2/3}(L\ln(2/\delta))^{1/3}\min\{\sqrt{KV}, \sqrt{L}\}}{V} \\
&\leq 18B^{1/3}T^{2/3}V^{-1/3}(L\ln(2/\delta))^{1/3}\min\{\sqrt{KV}, \sqrt{L}\},
\end{aligned}$$

where the last step uses $\tilde{V}^{2/3} \leq (5V)^{2/3} \leq 3V^{2/3}$. We now show that the term involving $V$ and the $\min\{\cdot\}$ is always bounded as follows:

**Claim 22.** $V^{-1/3}\min\{\sqrt{KV}, \sqrt{L}\} \leq K^{1/3}L^{1/6}$.

*Proof.* If $KV \leq L$, then $V \leq L/K$, and the expression equals $V^{-1/3}\sqrt{KV} = V^{1/6}\sqrt{K} \leq L^{1/6}K^{1/3}$. On the other hand, if $L \leq KV$, then $V \geq L/K$, and the expression is equal to $V^{-1/3}\sqrt{L} \leq K^{1/3}L^{1/6}$. $\qquad\square$

Thus, in this case, Term 1 is $O\left(T^{2/3}L^{1/2}(BK\log(2/\delta))^{1/3}\right)$.

Finally, assume that $\lambda_\star$ is the first term in its definition, i.e.,

$$\lambda_\star = 6L^2\ln(4LT/\delta) \geq (T\tilde{V}/B)^{2/3}(L\ln(2/\delta))^{1/3},$$

which implies

$$V \leq \tilde{V} \leq 6^{3/2}L^{5/2}\ln(4LT/\delta)^{3/2}(B/T)(\ln(2/\delta))^{-1/2}. \tag{32}$$

Thus, we have the bound

$$\text{Term } 1 \leq TB\sqrt{KV} = O\left(T^{1/2}B^{3/2}L^{5/4}K^{1/2}\log(LT/\delta)\right) = O_T(\sqrt{T}).$$

In summary, we have the bound,

$$\text{Term } 1 \leq \text{Term } 3 + O\left(T^{2/3}L^{1/2}(BK\log(2/\delta))^{1/3}\right) + O_T(\sqrt{T}). \tag{33}$$

**Term 2:** Plugging in the definition of $n_\star$ yields

$$\text{Term } 2 = n_\star B\sqrt{L} \leq T^{2/3}(K\ln(N/\delta))^{1/3}\max\{BL^{1/6}, B^{1/3}L^{-1/6}\}. \tag{34}$$

**Term 3:** Note that

$$\begin{aligned}
1/\sqrt{n_\star} &= T^{-1/3}(K\ln(N/\delta))^{-1/6}\min\{L^{1/6}, (BL)^{1/3}\}, \\
\lambda_\star &\geq 6L^2\ln(4LT/\delta) \geq L^2\ln(2/\delta),
\end{aligned}$$

so

$$\begin{aligned}
\text{Term } 3 &= 4T\left(\sqrt{\frac{cL\ln(2/\delta)}{\lambda_\star}} + B\right)\sqrt{\frac{K\ln(2N/\delta)}{n_\star}} \\
&\leq O\left(T^{2/3}\left(\frac{1}{\sqrt{L}} + B\right)(K\ln(N/\delta))^{1/3}\min\{L^{1/6}, (BL)^{1/3}\}\right).
\end{aligned}$$

Now if $B \geq \frac{1}{\sqrt{L}}$, then the min above is achieved by the $L^{1/6}$ term, so the bound is

$$O\left(T^{2/3}BL^{1/6}(K\log(N/\delta))^{1/3}\right).$$

If $B \leq \frac{1}{\sqrt{L}}$, then the min is achieved by the $(BL)^{1/3}$ term, so the bound is

$$O\left(T^{2/3}B^{1/3}L^{-1/6}(K\log(N/\delta))^{1/3}\right).$$

Thus,

$$\text{Term 3} = O\left(T^{2/3}(K\log(N/\delta))^{1/3}\left(BL^{1/6} + B^{1/3}L^{-1/6}\right)\right). \tag{35}$$

**Term 4:** We distinguish two cases. If $\lambda_\star = 6L^2\ln(4LT/\delta)$ then Eq. (32) holds and thus

$$V = O\left(L^{5/2}\ln(4LT/\delta)^{3/2}(B/T)\right).$$

We then have

$$\text{Term 4} = 2T\sqrt{\frac{cL\ln(2/\delta)}{\lambda_\star}}\min\{\sqrt{KV}, 2\sqrt{L}\}$$

$$\leq O\left(\frac{T}{\sqrt{L}}\sqrt{KV}\right) = O\left(\sqrt{BTK}L^{3/4}\ln(LT/\delta)\right) = O_T(\sqrt{T}).$$

Otherwise, $\lambda_\star = (T\tilde{V}/B)^{2/3}\left(L\ln(2/\delta)\right)^{1/3}$, and since $\tilde{V} \geq V$, we obtain

$$\text{Term 4} = 2T\sqrt{\frac{cL\ln(2/\delta)}{\lambda_\star}}\min\{\sqrt{KV}, 2\sqrt{L}\}$$

$$\leq O\left(T\sqrt{\frac{L\log(2/\delta)}{(TV/B)^{2/3}(L\log(2/\delta))^{1/3}}}\min\{\sqrt{KV}, \sqrt{L}\}\right)$$

$$= O\left(T^{2/3}B^{1/3}L^{1/3}V^{-1/3}(\log(2/\delta))^{1/3}\min\{\sqrt{KV}, \sqrt{L}\}\right)$$

$$= O\left(T^{2/3}(BK)^{1/3}L^{1/2}(\log(2/\delta))^{1/3}\right), \tag{36}$$

where the last step is Claim 22. This is the leading-order term since the other cases are $O_T(\sqrt{T})$.

**Putting everything together:** Combining Eqs. (33), (34), (35), and (36), we obtain the bound on the sum of the exploration and exploitation regret:

$$\text{Regret} = O\left(T^{2/3}(K\log(N/\delta))^{1/3}\max\{B^{1/3}L^{1/2}, BL^{1/6}\}\right) + O_T(\sqrt{T}).$$

## G  Deviation Bounds

Here, we collect several deviation bounds that we use in our proofs. All of these results are well known and we point to references rather than provide proofs. The first inequality, which is a Bernstein-type deviation bound for martingales, is Freedman's inequality, taken from Beygelzimer et. al [24]

**Lemma 23** (Freedman's Inequality). *Let $X_1, X_2, \ldots, X_T$ be a sequence of real-valued random variables. Assume for all $t \in \{1, 2, \ldots, T\}$ that $X_t \leq R$ and $\mathbb{E}[X_t|X_1, \ldots, X_{t-1}] = 0$. Define $S = \sum_{t=1}^T X_t$ and $V = \sum_{t=1}^T \mathbb{E}[X_t^2|X_1, \ldots, X_{t-1}]$. For any $\delta \in (0,1)$ and $\lambda \in [0, 1/R]$, with probability at least $1 - \delta$:*

$$S \leq (e-2)\lambda V + \frac{\ln(1/\delta)}{\lambda}$$

We also use Azuma's inequality, a Hoeffding-type inequality for martingales.

**Lemma 24** (Azuma's Inequality). *Let $X_1, X_2, \ldots, X_T$ be a sequence of real-valued random variables. Assume for all $t \in \{1, 2, \ldots, T\}$ that $X_t \leq R$ and $\mathbb{E}[X_t | X_1, \ldots, X_{t-1}] = 0$. Define $S = \sum_{t=1}^{T} X_t$. For any $\delta \in (0, 1)$, with probability at least $1 - \delta$:*

$$S \leq R\sqrt{2T \ln(1/\delta)}$$

We also make use of a vector-valued version of Hoeffding's inequality, known as the Hanson-Wright inequality, due to Rudelson and Vershynin [25].

**Lemma 25** (Hanson-Wright Inequality [25]). *Let $X = (X_1, \ldots, X_n)$ be a random vector with independent components satisfying $\mathbb{E}X_i = 0$ and $|X_i| \leq \kappa$ almost surely. There exists a universal constant $c_0 > 0$ such that, for any $A \in \mathbb{R}^{n \times n}$ and any $\delta \in (0, 1)$, with probability at least $1 - \delta$,*

$$|X^T A X - \mathbb{E}X^T A X| \leq \kappa^2 \sqrt{c_0 \|A\|_F^2 \log(2/\delta)} + c_0 \kappa^2 \|A\| \log(2/\delta),$$

*where $\| \cdot \|_F$ is the Frobenius norm and $\|\cdot\|$ is the spectral norm.*

Finally, we use a well known matrix-valued version of Bernstein's inequality, taken from Tropp [26].

**Lemma 26** (Matrix Bernstein). *Consider a finite sequence $\{X_k\}$ of independent, random, self-adjoint matrices with dimension $d$. Assume that for each random matrix we have $\mathbb{E}X_k = 0$ and $\lambda_{\max}(X_k) \leq R$ almost surely. Then for any $\delta \in (0, 1)$, with probability at least $1 - \delta$:*

$$\lambda_{\max}\left(\sum_k X_k\right) \leq \sqrt{2\sigma^2 \ln(d/\delta)} + \frac{2}{3}R \log(d/\delta) \qquad \text{with} \qquad \sigma^2 = \left\|\sum_k \mathbb{E}(X_k^2)\right\|,$$

*where $\|\cdot\|$ is the spectral norm.*

## Additional References

[24] A. Beygelzimer, J. Langford, L. Li, L. Reyzin, and R. E. Schapire. Contextual bandit algorithms with supervised learning guarantees. In *AISTATS*, 2011.

[25] M. Rudelson and R. Vershynin. Hanson-wright inequality and sub-gaussian concentration. *Electronic Communications in Probability*, 2013.

[26] J. A. Tropp. User-Friendly Tail Bounds for Sums of Random Matrices. *FOCM*, 2011.