[Reviews · NeurIPS 2016]

Reviewer 1

Summary

The authors consider a "semi-bandit" learning setting, in which a learner selects a group of L actions from a base set of K actions across a sequence of rounds. On each round, nature determines rewards for each base action, and reveals these values for the L base actions chosen by the learner (the semi-bandit feedback). The reward gained by the learner for the composite action is some weighted linear combination of the L base rewards, where this weighting might be unknown to the learner (and depends on the ordering of the base actions). The key contribution over previous work is to extend this type of problem to a contextual setting, where the goal is to compete with the best policy in some class. The authors assume access to a supervised learning oracle, which is able to return the best policy for some finite dataset of contexts, base-rewards, and weightings, and give an algorithm which achieves sqrt{T} regret while making a polynomial number of calls to this oracle.

Qualitative Assessment

The closest work to this setting can be found in Kale et al., which considers the same problem and gives similar rates. However, the algorithm is a modification of EXP4 and requires that the learning maintain a distribution over all policies in the policy class. In contrast, the authors use a technique very similar to that of Agarwal et. al. (https://arxiv.org/pdf/1402.0555v2.pdf) to solve a corresponding optimization problem via coordinate descent (and calls to the supervised learning oracle) while maintaining a distribution over policies with sparse support. The authors claim that the Kale et al. work also has the limitation that the weight vector must be w = 1. However, this is a somewhat dubious claim in the known-weights setting (unless I am mistaken, it seems that one could simply construct -w_l y_t(a_{t,l}) as the base-action loss for the Kale et al. algorithm). Overall, the big ideas of this paper are adaptations of the ideas in Agarwal et. al. However, the analysis is still quite involved and delicate, and ultimately the problem considered is an important one due to applications in online content recommendation.

Confidence in this Review

2-Confident (read it all; understood it all reasonably well)


Reviewer 2

Summary

The authors study the problem of contextual semi-bandits and efficient algorithms for this problem when a supervised learning oracle is provided. The authors consider the case when the reward is a known linear function of each individual feedbacks and also the case when the linear transformation is unknown. In the case when the linear transformation is known the algorithm proposed is similar to the one provided by Agarwal et al [1] but with some differences. In the case of unknown transformation the algorithm is a phased explore-exploit problem where the reward vector is estimated first and then used with an optimal policy. The paper is fairly well written.

Qualitative Assessment

Comments: 1) the OP problem is stated as a feasibility problem. Why not consider the optimization version where we try to find the policy with the smallest possible empirical regret subject to variance constraints (i.e optimize LHS in equation 4 subject to constraint 5). My guess is that the AMO is more suitable for solving the feasibility problem but it would be good to see it explicitly stated. 2) It would have been nice to see more discussion about how the AMO is used in the main body of the paper. For example let us say my policy class is linear classifiers then what does the AMO do?

Confidence in this Review

2-Confident (read it all; understood it all reasonably well)


Reviewer 3

Summary

This paper considers the contextual combinatorial semi-bandit problems, and proposes learning algorithms based on supervised learning oracles for both (1) the case with *known* weights and (2) the case with *unknown* weights. Regret bounds are developed for both cases (Theorem 1 and 2), and preliminary experiment results are demonstrated (Section 6).

Qualitative Assessment

This paper is very interesting in general, and I believe that it has met the standard of NIPS poster. In particular, to the best of my knowledge, this is the first paper considering contextual combinatorial semi-bandits with *unknown* weights. However, I think some parts of the paper can still be improved, and will appreciate it if the authors polish the final version of the paper accordingly: 1) In Theorem 2: the O(T^{2/3}) regret bound is somewhat unsatisfactory since I am expecting an O(T^{1/2}) regret bound. If the authors believe that the O(T^{2/3}) regret bound is intrinsic, please discuss. If the authors believe that it is due to unsatisfactory analysis, please also discuss (i.e. which step of the analysis leads to this non-tight regret bound). 2) Algorithm 2 is hard to digest. Please rewrite the motivation and explanation of the algorithm. 3) There is a recent paper on contextual combinatorial semi-bandit, which is directly relevant. The authors might cite it. @article{wen2014efficient, title={Efficient learning in large-scale combinatorial semi-bandits}, author={Wen, Zheng and Kveton, Branislav and Ashkan, Azin}, journal={http://jmlr.org/proceedings/papers/v37/wen15.html}, year={2014} }

Confidence in this Review

3-Expert (read the paper in detail, know the area, quite certain of my opinion)


Reviewer 4

Summary

The paper presented a new algorithm for solving contextual semibandit problems, that used an oracle, and presented proofs about the regret of the algorithm, as well as results comparing this algorithm against others on two data sets for search.

Qualitative Assessment

The contextual semibandits problem could be much better explained, it wasn't immediately clear on first reading, and required rereading and mapping the formal definitions to the examples. The problems tackled in this paper are very important, difficult, and have broad applications and relevance. In particular, tackling the case where the weights from the action list to the reward are unknown. The paper is a compelling combination of tackling a novel problem, comprehensive theoretical analysis, and convincing application to a practical data set. The authors were honestly and informatively even handed about outlining the insights as well as limitations of their proposed algorithm and results.

Confidence in this Review

1-Less confident (might not have understood significant parts)


Reviewer 5

Summary

Authors address an online learning problem where at each iteration the objective is to choose a list of items given the contextual information and based on which the learner receives some scalar feedback values for each individual selected items and a reward value: authors refer this problem as contextual semi-bandit problem in partial feedback setting, and proposed efficient algorithms (with sublinear regret guarantees) for this problem on different settings.

Qualitative Assessment

1. The problem formulation and contribution looks notable although the proposed algorithms are not intuitively well motivated. 2. The paper is difficult to follow, its not well written, there are various notational inconsistencies/ill defined quantities, e.g. definition of p_{min} is vague; in Algorithm 1 (VCEE), Q_t and \tilde{Q}_t are not properly defined. 3. Its not clear how the regret guarantee depends on the size of the policy set, authors mentioned that the dependence is O(log|\pi|) in the introduction but its not clear from Theorem 1 and 2. 4. It might be worth giving some insights on how this algorithm can be generalized for more general reward functions other than the linear structure which is currently addressed in the paper.

Confidence in this Review

2-Confident (read it all; understood it all reasonably well)


Reviewer 6

Summary

This paper tries to reduce contextual semibandits to supervised learning, so that powerful supervised learning methods can be leveraged in this partial-feedback setting. The authors claimed that the policies used to select arm in contextual semibandits is a large but constrained class. Such a class enables us to learn an expressive policy, but introduce a computational challenge of finding a good policy without direct enumeration. This paper build on the supervised learning literature, including logistic regression and SVMs for linear classifier and boosting for tree ensembles. They access the policy class exclusively through a supervised learning algorithm, viewed as oracle.

Qualitative Assessment

The oracle-based algorithm for contextual semibandits problems proposed in this paper is quite novel (as far as I know) and interesting. Both theoretical regret bound and empirical evaluation support their claims.

Confidence in this Review

1-Less confident (might not have understood significant parts)